# INRFlow: Flow Matching for INRs in Ambient Space

Yuyang Wang [1]   Anurag Ranjan [1 2]   Josh Susskind [1]   Miguel Angel Bautista [1]

## Abstract

Flow matching models have emerged as a powerful method for generative modeling on domains like images or videos, and even on irregular or unstructured data like 3D point clouds or even protein structures. These models are commonly trained in two stages: first, a data compressor is trained, and in a subsequent training stage a flow matching generative model is trained in the latent space of the data compressor. This two-stage paradigm sets obstacles for unifying models across data domains, as hand-crafted compressors architectures are used for different data modalities. To this end, we introduce **INRFlow**, a domain-agnostic approach to learn flow matching transformers directly in ambient space. Drawing inspiration from INRs, we introduce a conditionally independent point-wise training objective that enables INRFlow to make predictions continuously in coordinate space. Our empirical results demonstrate that INRFlow effectively handles different data modalities such as images, 3D point clouds and protein structure data, achieving strong performance in different domains and outperforming comparable approaches. INRFlow is a promising step towards domain-agnostic flow matching generative models that can be trivially adopted in different data domains.

## 1. Introduction

Recent advances in generative modeling have enabled learning complex data distributions by combining both powerful architectures and training objectives. In particular, state-of-the-art approaches for image (Esser et al., 2024), video (Dai et al., 2023) or 3D point cloud (Vahdat et al., 2022) generation are based on the concept of iteratively transforming data into Gaussian noise. Diffusion models were originally proposed following this idea and pushing the quality of generated samples in many different domains, including images (Dai et al., 2023; Rombach et al., 2022; Preechakul et al., 2022), 3D point clouds (Luo & Hu, 2021), graphs (Hoogeboom et al., 2022) and video (Ho et al., 2022a). More recently, flow matching (Lipman et al., 2023) and stochastic interpolants (Ma et al., 2024) have been proposed as generalized formulations of the noising process, moving from stochastic gaussian diffusion processes to general paths connecting a base (*e.g.* Gaussian) and a target (*e.g.* data) distribution.

In practice, these iterative refinement approaches are commonly applied in a *latent space* obtained from a pre-trained compressor model. Thus, the training recipe consists of two independent training stages: *compressor* (auto-encoder) training and subsequent *generative modeling* in latent space. General purpose transformer architectures have been used in the generative modeling step in latent space (Peebles & Xie, 2023; Ma et al., 2024; Esser et al., 2024). However, the first stage compressor uses architectures that are specific to the data domain, requiring hand-crafted inductive biases (*i.e.* ConvNets for image data (Rombach et al., 2022), PointNet for point clouds (Vahdat et al., 2022), Evoformer for protein structures (Jumper et al., 2021)).

We see this as one of the core issues preventing the ML community to develop truly domain-agnostic generative models that can be applied to different data domains in a trivial manner. Our goal in this paper is to provide a powerful single training stage approach that is domain-agnostic and simple to implement in practice, thus dispensing with the complexities of two-stage training recipes and enabling modeling of different data modalities directly in ambient (*i.e.* data) space.

It is worth noting that training diffusion or flow matching models in ambient space is indeed possible when using domain specific architecture designs and training recipes. In the image domain, approaches have exploited its dense nature and applied cascaded U-Nets (Ho et al., 2021; 2022b), joint training of U-Nets at multiple resolutions (Gu et al., 2023), multi-scale losses (Hoogeboom et al., 2023) or U-Net transformer hybrids architectures (Crowson et al., 2024), obtaining strong results. However, developing strong domain-agnostic models, using general purposes architectures that

---

[1] Apple, Machine Learning Research [2] Work done while at Apple. Correspondence to: Yuyang Wang <yuyangw@apple.com>, Miguel Angel Bautista <mbautistamartin@apple.com>.

*Proceedings of the $42^{nd}$ International Conference on Machine Learning*, Vancouver, Canada. PMLR 267, 2025. Copyright 2025 by the author(s).

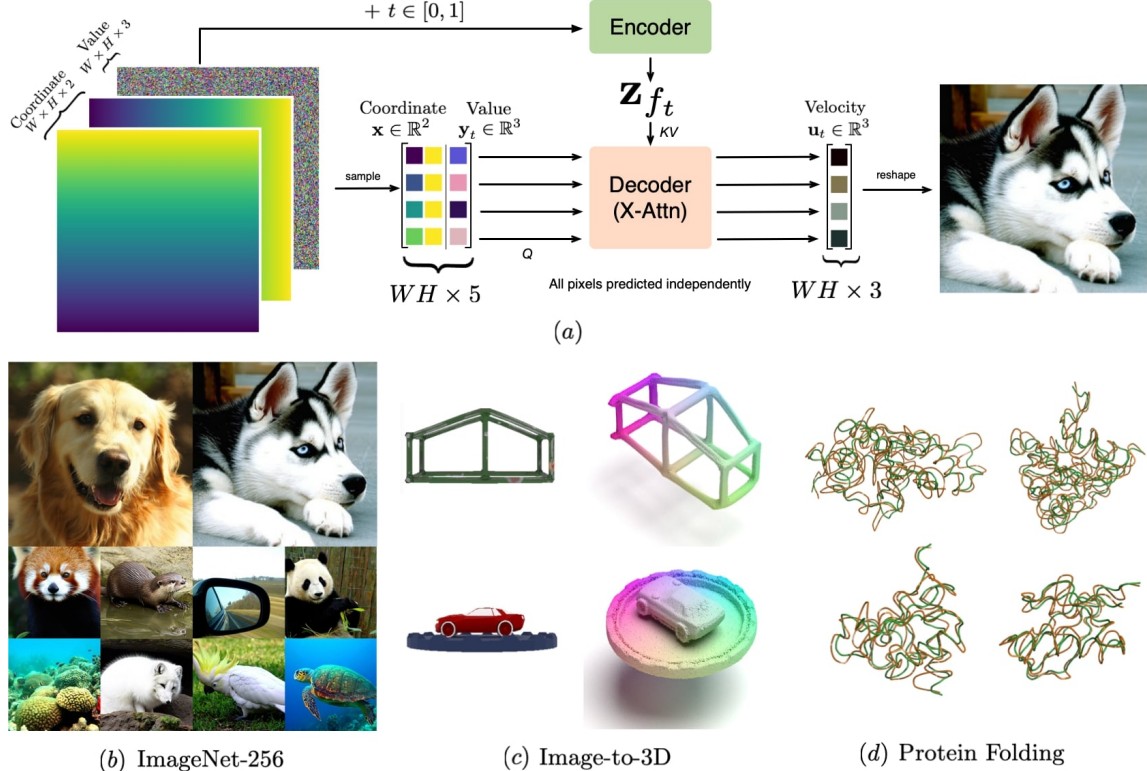

*Figure 1.* (a) High level overview of INRFlow using the image domain as an example. Our model can be interpreted as an encoder-decoder model where the decoder makes predictions independently for each coordinate-value pair given $z_{f_t}$. For different data domains, the coordinate and value dimensionality changes, but the model is kept the same. (b) Samples generated by INRFlow trained on ImageNet 256×256. (c) Image-to-3D point clouds generated by training INRFlow on Objaverse (Deitke et al., 2023). (d) Protein structures generated by INRFlow trained on SwissProt (Boeckmann et al., 2003). GT protein structures are depicted in green while the generated structures by INRFlow are show in orange.

can be applied across different data domains remains an important open problem.

In this paper, we answer a three part question: *Can we learn flow matching models in **ambient space**, in a **single training stage** and using a **domain agnostic** architecture?* Our goal is to unify different data domains under the same training recipe. To achieve this, we introduce INRFlow (INRFlow), see Fig. 1(a). INRFlow makes progress towards the goal of unifying flow matching generative modeling across data domains. Drawing inspiration from INRs, we formulate a conditionally independent point-wise training objective that enables training directly in ambient space and can be densely (*e.g.* continuously) evaluated during inference. In the image domain, this means that INRFlow models the probability of a pixel value given its coordinate (*i.e. the probabilistic extension of an INR*), allowing to generate images at different resolution than the one used during training (see Fig. 4(a)). We show generated samples from INRFlow trained on ImageNet-256 in Fig. 1(b), image-to-3D on Objaverse in Fig. 1(c) and protein structures on SwissProt Fig. 1(d) (see additional samples in Fig. 9, 12,

13). Our contributions are summarized as follows:

- We propose INRFlow, a flow matching generative transformer that works on ambient space to enable single stage generative modeling on different data domains.

- Our results show that INRFlow, though domain-agnostic, achieves competitive performance on image and 3D point cloud generation compared with strong domain-specific baselines.

- Our point-wise training objective allows for efficient training via sub-sampling dense domains like images while also enabling resolution changes at inference time.

## 2. Related Work

Diffusion models have been the major catalyzer of progress in generative modeling, these approaches learn to reverse a forward process that gradually adds Gaussian noise to corrupt data samples (Ho et al., 2020). Diffusion models

are notable for their simple and robust training objective. Extensive research has explored various formulations of the forward and backward processes (Song et al., 2021; Rissanen et al., 2022; Bansal et al., 2022), particularly in the image domain. In addition, different denoising networks have been proposed for different data domains like images (Nichol & Dhariwal, 2021), videos (Ho et al., 2022a), and geometric data (Luo & Hu, 2021). More recently, flow matching (Liu et al., 2023; Lipman et al., 2023) and stochastic interpolants (Ma et al., 2024) have emerged as flexible formulations that generalized Gaussian diffusion paths, allowing to define different paths to connect a base and a target distribution. These types of models have shown incredible results in the image domain (Ma et al., 2024; Esser et al., 2024) when coupled with transformer architectures (Vaswani et al., 2017) to model distributions in latent space learnt by data compressors (Peebles & Xie, 2023; Ma et al., 2024; Rombach et al., 2022; Vahdat et al., 2022; Zheng et al., 2023; Gao et al., 2023). Note that these data compressors use domain specific architectures with hand-crafted inductive biases.

In an attempt to unify generative modeling across various data domains, continuous data representations (also referred to as *implicit neural representation*, *neural fields* or *neural operators*) have shown potential in different approaches: From Data to Functa (Functa) (Dupont et al., 2022a), Generative Manifold Learning (GEM) (Du et al., 2021a), and Generative Adversarial Stochastic Process (GASP) (Dupont et al., 2022b) have studied the problem of generating continuous representations of data. More recently Infinite Diffusion (Bond-Taylor & Willcocks, 2023) and PolyINR (Singh et al., 2023) have shown great results in the image domain by modeling images as continuous functions. However, both of these approaches make strong assumptions about image data. In particular, (Bond-Taylor & Willcocks, 2023) interpolates sparse pixels to an euclidean grid to then process it with a U-Net. On the other hand, (Singh et al., 2023) uses a patching and 2D convolution in the discriminator. Our approach also relates to DPF (Zhuang et al., 2023), a diffusion model that acts on function coordinates and can be applied in different data domains on a grid at low resolutions (*i.e.* 64×64). Our approach is able to deal with higher resolution functions (*e.g.* 256x256 vs. 64x64 resolution images) on large scale datasets like ImageNet, while also tackling unstructured data domains that do not live on an Euclidean grid (*e.g.* like 3D point clouds and protein structures).

# 3. Method

## 3.1. Data as Continuous Coordinate → Value Maps

We interpret our empirical data distribution $q$ to be composed of maps $f \sim q(f)$. These maps take *coordinates* $\boldsymbol{x}$ as input to *values* $\boldsymbol{y}$ as output. For images, maps are defined

from 2D pixel coordinates $\boldsymbol{x} \in \mathbb{R}^2$ to corresponding RGB values $\boldsymbol{y} \in \mathbb{R}^3$, thus $f : \mathbb{R}^2 \to \mathbb{R}^3$, where each image is a different map. For 3D point clouds, $f$ can be interpreted as a deformation that maps coordinates from a fixed base configuration in 3D space to a deformation value also in 3D space, $f : \mathbb{R}^3 \to \mathbb{R}^3$, as in the image case, each 3D point cloud corresponds to a different deformation map $f$. For ease of notation, we define coordinates $\boldsymbol{x}$ and values $\boldsymbol{y}$ of any given map $f$ as $\boldsymbol{x}_f$ and $\boldsymbol{y}_f$, respectively. Fig. 1(a) shows an example of such maps in the image domain.

In practice, analytical forms for these maps $f$ are unknown. In addition, different from previous approaches (Dupont et al., 2022a; Du et al., 2021a), we do not fit separate INRs to each data sample via reconstruction, since that would involve a separate training stage fitting an MLP for each map (Dupont et al., 2022a; Bauer et al., 2023; Du et al., 2021a). As a result, we assume we are only given sets of corresponding *coordinate* and *value* pairs resulting from observing these maps at a particular sampling rate (*e.g.* at a particular resolution in the image case). In the following, we develop an end-to-end approach that can directly take these coordinate-value sets as training data and train a model that extends INRs to the probabilistic setting.

## 3.2. Flow Matching and Stochastic Interpolants

We consider generative models that learn to reverse a time-dependent forward process that turns data samples (*i.e.* maps $f$ in our case) $f \sim q(f)$ into noise $\epsilon \sim \mathcal{N}(0, \mathbf{I})$.

$$f_t = \alpha_t f + \sigma_t \epsilon \qquad (1)$$

Both flow matching (Lipman et al., 2023) and stochastic interpolant (Ma et al., 2024) formulations build this forward process in Eq. 1 so that it interpolates exactly between data samples $f$ at time $t = 0$ and $\epsilon$ at time $t = 1$, with $t \in [0, 1]$. In particular, $p_1(f) \sim \mathcal{N}(0, \mathbf{I})$ and $p_0(f) \approx q(f)$. In this case, the marginal probability distribution $p_t(f)$ of $f$ is equivalent to the distribution of the probability flow ODE with the following velocity field (Ma et al., 2024):

$$d_t f_t = \boldsymbol{u}_t(f_t) d_t \qquad (2)$$

where the velocity field is given by the following conditional expectation,

$$\boldsymbol{u}_t(f) = \mathbb{E}[d_t f_t | f_t = f] = \\ = d_t \alpha_t \mathbb{E}[f_0 | f_t = f] + d_t \sigma_t \mathbb{E}[\epsilon | f_t = f]. \qquad (3)$$

Under this formulation, samples $f_0 \sim p_0(f)$ are generated by solving the probability flow ODE in Eq. 2 backwards in time (*e.g.* . flowing from $t = 1$ to $t = 0$), where

$p_0(f) \approx q(f)$. Note that both the flow matching (Lipman et al., 2023) and stochastic interpolant (Ma et al., 2024) formulations decouple the time-dependent process formulation from the specific choice of parameters $\alpha_t$ and $\sigma_t$, allowing for more flexibility. Throughout the presentation of our method we will assume a rectified flow (Liu et al., 2023; Lipman et al., 2023) or linear interpolant path (Ma et al., 2024) between noise and data, which define a straight path to connect data and noise: $f_t = (1 - t)f_0 + t\epsilon$. Note that our framework for learning flow matching models for coordinate-value sets can be used with any path definition. Compared with diffusion models (Ho et al., 2020), linear flow matching objectives result in better training stability and more modeling flexibility (Ma et al., 2024; Esser et al., 2024) which we observed in our early experiments.

### 3.3. INRFlow

We now turn to the task of formulating a flow matching training objective for data distributions of maps $f$. We recall that in practice we do not have access to an analytical or parametric form for these maps $f$ (*e.g.* we do not requite to pretrain INRs for each training sample), and we are only given sets of corresponding *coordinate* $\boldsymbol{x}_f$ and *value* $\boldsymbol{y}_f$ pairs resulting from observing the mapping at a particular rate. As a result, we need to formulate a training objective that can take these sets of coordinate-value as training data.

In order to achieve this, we first observe that the target velocity field $\boldsymbol{u}_t(f_t)d_t$ can be decomposed across both the domain and co-domain of $f_t$, resulting in a *point-wise velocity field* $\boldsymbol{u}_t(\boldsymbol{x}_{f_t}, \boldsymbol{y}_{f_t})d_t$, defined for corresponding coordinate and value pairs of $f_t$. As an illustrative example in the image domain, this means that the *target velocity field can be independently evaluated* for any pixel coordinate $\boldsymbol{x}_{f_t}$ with corresponding value $\boldsymbol{y}_{f_t}$, so that $\boldsymbol{u}_t(\boldsymbol{x}_{f_t}, \boldsymbol{y}_{f_t}) \in \mathbb{R}^3$. Note that one can always decompose target velocity fields in this way since the time-dependent forward process in Eq. 1 aggregates data and noise *independently* (*e.g.* point-wise) across the domain of $f$. Again, using the image domain as an example, the time-dependent forward process of a pixel at coordinate $\boldsymbol{x}_f$ is not dependent on other pixel positions or values.

Our goal now is to formulate a training objective to match this point-wise independent velocity field. We want our neural network $\boldsymbol{v}_\theta$ parametrizing the velocity field to be able to independently predict a velocity for any given coordinate and value pair $\boldsymbol{x}_{f_t}$ and $\boldsymbol{y}_{f_t}$. However, this point-wise independent prediction is futile without access to additional contextual conditioning information about the underlying function $f_t$ at time $t$. This is because even if the forward process is point-wise independent, real data exhibits strong dependencies across the domain $f$ that need to be captured by the model. For example, in the image domain, pixels

are not independent from each other and natural images show strong both short and long spatial dependencies across pixels. In order to solve this, we introduce a latent variable $\boldsymbol{z}_{f_t}$ that encodes contextual information from a set of given coordinate and value pairs of $f_t$. This contextual latent variable allows us to formulate the learnt velocity field to be *conditionally independent* for coordinate-value pairs given $\boldsymbol{z}_{f_t}$. The final point-wise conditionally independent CFM loss, which we denote as CICFM loss is defined as:

$$\mathbb{E}_{t \sim \mathcal{U}[0,1], f \sim q(f)} ||\boldsymbol{v}_\theta(\boldsymbol{x}_{f_t}, \boldsymbol{y}_{f_t}, t|\boldsymbol{z}_{f_t}) - \boldsymbol{u}_t(\boldsymbol{x}_f, \boldsymbol{y}_f|\epsilon)||_2^2, \tag{4}$$

where the target velocity field $\boldsymbol{u}_t(\boldsymbol{x}, \boldsymbol{y}|\epsilon)$ is defined as a rectified flow (Liu et al., 2023; Lipman et al., 2023; Ma et al., 2024): $\boldsymbol{u}_t(\boldsymbol{x}_f, \boldsymbol{y}_f|\epsilon) = \frac{\epsilon - \boldsymbol{y}_{f_t}}{1 - t}$.

One of the core challenges of learning this type of generative models is obtaining a latent variable $\boldsymbol{z}_{f_t}$ that effectively captures intricate dependencies across the domain of the function, specially for high resolution stimuli like images. In particular, the architectural design decisions are extremely important to ensure that $\boldsymbol{z}_{f_t}$ does not become a bottleneck during training. In the following we review our proposed architecture.

### 3.4. Network Architecture

We base our model on the general PerceiverIO design (Jaegle et al., 2022), Fig. 2 illustrates the architectural pipeline of INRFlow. At a high level, our encoder network takes a set of coordinate-value pairs and encodes them to learnable latents through cross-attention. These latents are then updated through several self-attention blocks to provide the final latents $\boldsymbol{z}_{f_t} \in \mathbb{R}^{L \times D}$. To decode the velocity field for a given coordinate-value pair we perform cross attention to $\boldsymbol{z}_{f_t}$, generating the final point-wise prediction for the velocity field $\boldsymbol{v}_\theta(\boldsymbol{x}_{f_t}, \boldsymbol{y}_{f_t}, t|\boldsymbol{z}_{f_t})$.

The encoder of a vanilla PerceiverIO relies solely on cross-attention to the latents $z_{f_t} \in \mathbb{R}^{L \times D}$ to learn spatial connectivity patterns between input and output elements, which we found to introduce a strong bottleneck during training. To ameliorate this, we make a key modifications to boost the performance. Firstly, our encoder utilizes spatial aware latents where each latent is assigned a "pseudo" coordinate. Coordinate-value pairs are assigned to latents based on their distances on coordinate space. During encoding, coordinate-value pairs interact with their assigned latents through cross-attention, this means that each of the $L$ latents only attends to a set of neighboring coordinate-value pairs. Latent vectors are then updated using several self-attention blocks. These changes in the encoder allow the model to effectively utilize spatial information while also saving compute when encoding large coordinate-value sets on ambient

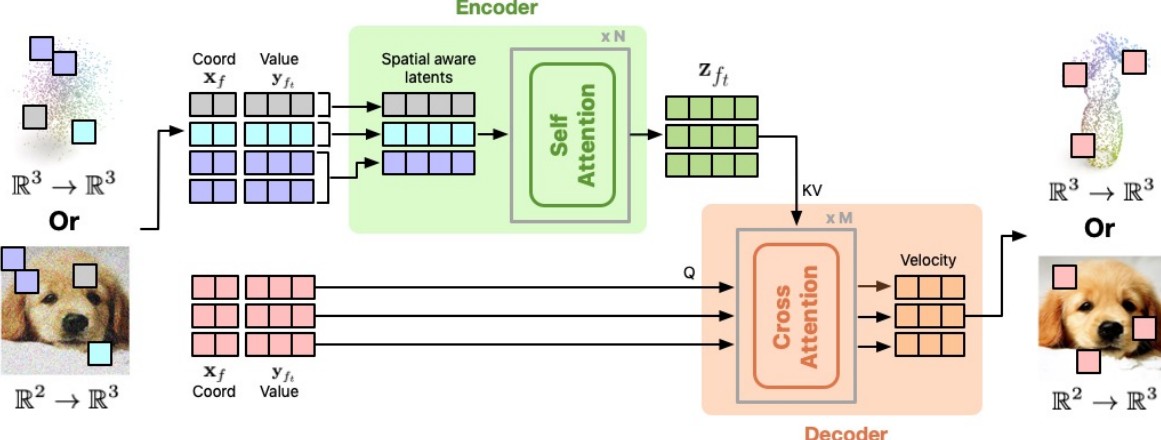

*Figure 2.* Architecture of our proposed INRFlow for different data domains including images and 3D point clouds. Note that models are trained for each data domain separately. Each spatial aware latent takes in a subset of neighboring context coordinate-value sets in coordinate space. The latents are then updated through self-attention. Decoded coordinate-value pairs cross attend to the updated latents $\boldsymbol{z}_{f_t}$ to decode the corresponding velocity.

space.

## 4. Experiments

We evaluate INRFlow on two challenging problems: image generation (FFHQ-256 (Karras et al., 2019), LSUN-Church-256 (Yu et al., 2015), ImageNet-128/256 (Russakovsky et al., 2015)), image-to-3D point cloud generation (Objaverse (Deitke et al., 2023)) and protein folding (SwissProt (Boeckmann et al., 2003)). Note that we use the same training recipe for all tasks, adapted for changes in coordinate-value pair dimensions in different domains. See App. A for more implementation details and training settings.

### 4.1. Image Generation

Given that INRFlow is an probabilistic extension of INRs we compare it with other generative models of the same type, namely approaches that operate in continuous function spaces. Tab. 1 shows a comparison of different image domain specific, as well as, function space models (*e.g.* approaches that model infinite-dimensional signals). INRFlow surpasses other generative models in function space on both FFHQ (Karras et al., 2019) and LSUN-Church (Yu et al., 2015) at resolution $256 \times 256$. Compared with generative models designed specifically for images, INRFlow also achieves comparable or better performance. When scaling up the model size, INRFlow-L demonstrates better performance than all the baselines on FFHQ-256 and Church-256, indicating that INRFlow can benefit from increasing model sizes.

We also evaluate the performance of INRFlow on large scale settings previously untapped for domain agnostic ap-

| Model | FFHQ-256 | Church-256 |
|---|---|---|
| *Domain specific models* | | |
| CIPS (Anokhin et al., 2021) | 5.29 | 10.80 |
| StyleSwin (Zhang et al., 2022) | 3.25 | 8.28 |
| UT (Bond-Taylor et al., 2022) | 3.05 | 5.52 |
| StyleGAN2 (Karras et al., 2020) | 2.35 | 6.21 |
| *Function space models* | | |
| GEM (Du et al., 2021b) | 35.62 | 87.57 |
| GASP (Dupont et al., 2022c) | 24.37 | 37.46 |
| ∞-Diff (Bond-Taylor & Willcocks, 2023) | 3.87 | 10.36 |
| **INRFlow-B** (ours) | 2.46 | 7.11 |
| **INRFlow-L** (ours) | **2.18** | **5.51** |

*Table 1.* FID$_{\text{CLIP}}$ (Kynkäänniemi et al., 2023) results for state-of-the-art function space approaches.

proaches, training INRFlow on ImageNet at both $128 \times 128$ and $256 \times 256$ resolutions. On ImageNet-128, shown in Tab. 2, INRFlow achieves an FID of 2.73, which is a a competitive performance in comparison to diffusion or flow-based generative baselines including ADM (Dhariwal & Nichol, 2021), CDM (Ho et al., 2021), and RIN (Jabri et al., 2023) which use domain-specific architectures for image generation. Besides, comparing to PolyINR (Singh et al., 2023) which also operates on function space, INRFlow achieves competitive FID, while obtaining better IS, precision and recall. In addition, we report results of INRFlow for ImageNet-256 on Tab. 3. We observed that INRFlow is slightly outperformed by latent space models like DiT (Peebles & Xie, 2023) and SiT (Ma et al., 2024). We highlight that these baselines rely on a pre-trained VAE compressor that was trained on datasets (*i.e.* 9.29M images) that are much larger than ImageNet, while INRFlow was trained only with ImageNet data. In addition, INRFlow achieves better performance than many of the baselines trained only with ImageNet data including ADM (Dhariwal & Nichol,

**Class-Conditional ImageNet 128x128**

| Model | FID↓ | IS↑ | Prec↑ | Rec↑ |
|---|---|---|---|---|
| *Adversarial models* | | | | |
| BigGAN-deep (Brock et al., 2019) | 6.02 | 145.8 | **0.86** | 0.35 |
| PolyINR (Singh et al., 2023) | **2.08** | 179.0 | 0.70 | 0.45 |
| *Diffusion models* | | | | |
| CDM (w/ cfg) (Ho et al., 2021) | 3.52 | 128.0 | - | - |
| ADM (w/ cfg) (Dhariwal & Nichol, 2021) | 2.97 | 141.3 | 0.78 | **0.59** |
| RIN (Jabri et al., 2023) | 2.75 | 144.0 | - | - |
| **INRFlow**-XL (ours) (cfg=1.5) | 2.73 | **187.6** | 0.80 | 0.58 |

*Table 2.* Benchmarking class-conditional image generation on ImageNet 128x128.

2021), CDM (Ho et al., 2021) and Simple Diffusion (U-Net) (Hoogeboom et al., 2023) which all use CNN-based architectures specific for image generation. Note that this is consistent with the results show in Tab. 1, where INRFlow outperforms all function space approaches.

When comparing with approaches using transformer architectures we find that INRFlow obtains performance comparable to RIN (Jabri et al., 2022) and HDiT (Crowson et al., 2024), with slightly worse FID and slightly better IS. However, INRFlow is a domain-agnostic architecture that can be trivially applied to different data domains like 3D point clouds or protein structure data (see Sect. 4.2 and Sect. 4.3). For completeness, we also include billion scale U-Net transformer hybrid models, Simple Diffusion (U-ViT 2B) and VDM++ (U-ViT 2B). We highlight that the simplicity of implementing and training INRFlow models in practice, and the trivial extension to different data domains are strong arguments favoring INRFlow. Finally, comparing with *e.g.* PolyINR (Singh et al., 2023) which is also a function space generative model we also find comparable performance, with slight worse FID but better Precision and Recall. It is worth noting that (Singh et al., 2023) applies a pre-trained DeiT model as the discriminator (Singh et al., 2023). Whereas our INRFlow makes no such assumption about the function or pre-trained models, enabling to trivially apply INRFlow to other domains like 3D point clouds (see Sect. 4.2) or protein folding.

### 4.2. Image-to-3D

We also showcase that INRFlow can directly integrate conditional information like images. We train an image-to-point-cloud INRFlow model on Objaverse (Deitke et al., 2023), which contains 800k 3D objects of wide variety, to illustrate the capability of INRFlow on large-scale 3D generative tasks. In particular, conditional information (i.e., an image) is integrated to our model through cross-attention. We train INRFlow with batch size 384 for 500k iterations. During sampling, we use an Euler-Maruyama sampler (Ma et al., 2024) with 500 steps to generate point clouds (see App. A.2 for more details on architecture and objaverse data

generation).

Tab. 4 shows the performance of INRFlow in comparison with recent baselines on Objaverse. We report ULIP-I (Xue et al., 2024) and P-FID (Nichol et al., 2022) following CLAY (Zhang et al., 2024). PointNet++ (Qi et al., 2017a;b; Nichol et al., 2022) is employed to evaluate P-FID. ULIP-I is an analogy to CLIP for text-to-image generation. ULIP-I is measured as the cosine similarity between point-cloud features from ULIP-2 model (Xue et al., 2024) and image features from CLIP model (Radford et al., 2021). The performance numbers of baseline models are directly borrowed from CLAY (Zhang et al., 2024). We calculate the metrics of our INRFlow on 10k sampled point clouds. In our case, P-FID is measured on point clouds with 4096 points following Shape-E (Jun & Nichol, 2023) while ULIP-I is measured on point clouds with 10k points following ULIP-2 (Xue et al., 2024). Note that since CLAY (Zhang et al., 2024) is not open-source, we do not have the access to the exact evaluation setting or the conditional images rendered from Objaverse. But all evaluation settings of INRFlow are provided for reproduction purpose. As shown in Tab. 4, our INRFlow achieves strong performance on image-to-3D generative tasks. Compared to CLAY (Zhang et al., 2024), which is a 2-stage latent diffusion model, INRFlow demonstrates very strong performance on both ULIP-I and P-FID.

Fig. 4(b) show examples of sampled point clouds and corresponding conditional images. As discussed in §4.4, INRFlow trained on Objaverse also enjoys the flexibility of resolution agnostic generation. In the App, Fig. 12 we show additional results sampled with more points than what the model was trained on. As shown, INRFlow learns to generate 3D objects with rich details that match the conditional images ultimately being able to generate a continuous surface.

### 4.3. Protein Folding

We now showcase the domain-agnostic prowess of INRFlow applying it to the protein folding problem (Jumper et al., 2021). From an ML perspective, this problem is a conditional 3D generation problem where we are given the amino-acid sequence (*e.g.* a sequence of discrete symbols from a vocabulary of 20 possible amino-acids) and we need to generate a 3D coordinate for each atom in the protein, where different amino-acids can have different numbers of atoms. In our experiments we use SwissProt set (Boeckmann et al., 2003) taking the ground truth structures from the AlphaFold Database (Varadi et al., 2022).

We adapt the coordinate and signal in INRFlow so that the coordinate becomes a feature representation of each atom in the protein and the signal represents the 3D point for that particular atom. This is similar to recent work on protein folding and conformer generation, which directly predicts

| Class-Conditional ImageNet 256x256 | | | | | | | | |
|---|---|---|---|---|---|---|---|---|
| Model | Agnostic | # Tr. Samples | # Params | bs × it. | FID↓ | IS↑ | Precision↑ | Recall↑ |
| *Adversarial models* | | | | | | | | |
| BigGAN-deep (Brock et al., 2019) | ✗ | 1.28M | - | - | 6.95 | 171.4 | **0.87** | 0.28 |
| PolyINR (Singh et al., 2023) | ✗ | 1.28M | - | - | 2.86 | 241.4 | 0.71 | 0.39 |
| *Latent space with pretrained VAE* | | | | | | | | |
| DiT-XL (cfg=1.5) (Peebles & Xie, 2023) | ✗ | **9.23M** | 675M | - | 2.27 | **278.2** | 0.83 | 0.57 |
| SiT-XL (cfg=1.5, SDE) (Ma et al., 2024) | ✗ | **9.23M** | 675M | **1.8B** | 2.06 | 270.2 | 0.82 | 0.59 |
| *Ambient space* | | | | | | | | |
| ADM (Dhariwal & Nichol, 2021) | ✗ | 1.28M | 554M | 507M | 10.94 | 100.9 | 0.69 | **0.63** |
| CDM (Ho et al., 2021) | ✗ | 1.28M | - | - | 4.88 | 158.7 | - | - |
| Simple Diff. (U-Net) (Hoogeboom et al., 2023) | ✗ | 1.28M | - | - | 3.76 | 171.6 | - | - |
| RIN (Jabri et al., 2023) | ✗ | 1.28M | 410M | 614M | 3.42 | 182.0 | - | - |
| HDiT (cfg=1.3) (Crowson et al., 2024) | ✗ | 1.28M | 557M | 742M | 3.21 | 220.6 | - | - |
| Simple Diff. (U-ViT) (Hoogeboom et al., 2023) | ✗ | 1.28M | **2B** | 1.4B | 2.77 | 211.8 | - | - |
| VDM++ (U-ViT) (Kingma & Gao, 2023) | ✗ | 1.28M | **2B** | 1.4B | 2.12 | 267.7 | - | - |
| **INRFlow**-XL (ours) (cfg=1.5) | ✓ | 1.28M | 733M | 780M | 3.74 | 228.8 | 0.82 | 0.52 |

*Table 3.* Top performing models for class-conditional image generation on ImageNet 256x256.

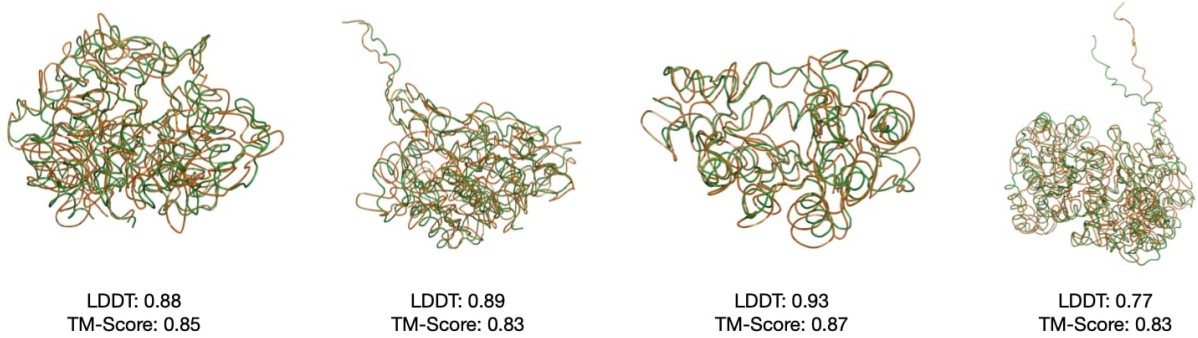

LDDT: 0.88    LDDT: 0.89    LDDT: 0.93    LDDT: 0.77
TM-Score: 0.85    TM-Score: 0.83    TM-Score: 0.87    TM-Score: 0.83

*Figure 3.* Examples of protein structures predicted by INRFlow on SwissProt, together with their LDDT and TM scores. The GT structures are depicted in green while the generated structures are show in orange. INRFlow accurately captures the global spatial distribution of protein backbones generating reasonable 3D structures for different protein sequences.

| Model | ULIP-I ↑ | P-FID ↓ |
|---|---|---|
| Shap-E (Jun & Nichol, 2023) | 0.1307 | - |
| Michelangelo (Zhao et al., 2024) | 0.1899 | - |
| CLAY (Zhang et al., 2024) | 0.2066 | 0.9946 |
| INRFlow (ours) | **0.2976** | **0.3638** |

*Table 4.* Image-conditioned 3D point cloud generation performance on Objaverse.

atom coordinates as opposed to frames of reference (Wang et al.; Abramson et al., 2024). For the spatially-aware latents we aggregate information from all the atoms corresponding to the same amino-acid into a single latent vector (see App. A.3 for more details on architecture and training recipe).

We show qualitative results on Fig. 3 and additional results on Fig. 13. INRFlow accurately captures the global distribution of protein backbones generating reasonable 3D structures for different protein sequences. These initial results are very encouraging and open up exciting design

spaces for future work developing protein folding network architectures that can leverage general purpose transformer blocks.

### 4.4. Resolution Agnostic Generation

An interesting property of INRFlow is that it decodes each coordinate-value pair independently, allowing resolution changes of resolution for generation. At inference time the user can define as many coordinate-value pairs as desired where the initial value of each pair at $t = 1$ is drawn from a Gaussian distribution. We now quantitatively evaluate the performance of INRFlow in this setting. In Tab. 5 we compare the FID of different recipes. First, INRFlow is trained on FFHQ-256 and bilinear or bicubic interpolation is applied to the generated samples to get images at 512. On the other hand, INRFlow can directly generate images at resolution 512 by simply increasing the number of coordinate-value pairs during inference without further tuning. As shown in Tab. 5 , INRFlow achieves lower FID

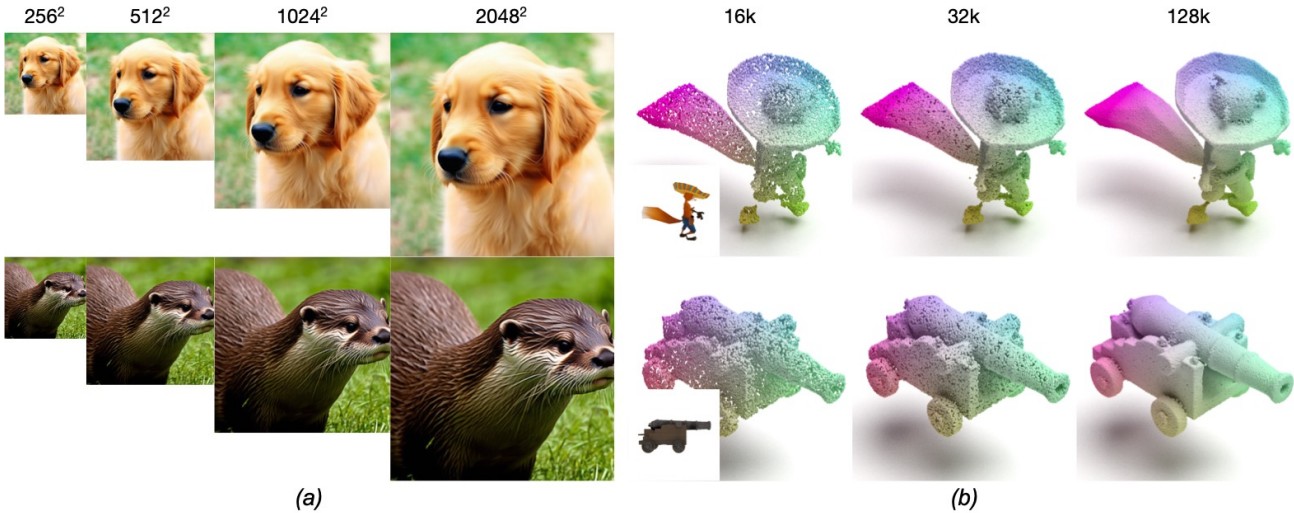

*Figure 4.* Examples of resolution agnostic generation for INRFlow models trained on ImageNet-256 (a), and Objaverse-16k in (b). To generate samples at higher resolutions than the one in training we fix the initial noise seed and increase the number of coordinate-value pairs evaluated by the model. Even though INRFlow was only train with samples at a fixed resolution (256 for ImageNet and 16k for Objaverse), it can still generate realistic samples at higher resolutions. These results show that INRFlow is learning a continuous probability density field.

when compared with other manually designed interpolation methods, showcasing the benefit of developing generative models on ambient space.

|  | INRFlow | Bilinear | Bicubic |
|---|---|---|---|
| FID($\downarrow$) | 23.09 | 35.05 | 24.34 |

*Table 5.* FID of different super resolution methods to generate images at resolution $512 \times 512$ for INRFlow trained on FFHQ-256.

We show examples of resolution agnostic generation for INRFlow models trained on ImageNet-256 Fig. 4(a) and Objaverse-16k in Fig. 4(b). Even though INRFlow was only train with samples at a fixed resolution (256 for ImageNet and 16k for Objaverse), it can still generate realistic samples at higher resolutions. For example, Fig. 4(a) shows samples generated from INRFlow trained on ImageNet-256 and sampled at resolutions up to 2k (see additional examples in Fig. 14). Fig. 4(b) shows point cloud with up to 128k points from INRFlow trained on Objaverse with only 16k points points per sample. It is worth noting that in INRFlow there's no cascading (Ho et al., 2022b) or multiple resolution training (Gu et al., 2023), we simply fix the initial noise seed and increase number of coordinate-value pairs that are evaluated. These results show that INRFlow is not trivially overfitting to the coordinate-value pairs in the training but rather learning a continuous probability density field in space from which an infinite number of points could be sampled. Generally speaking, this also provides the potential to efficiently train flow matching generative models without the need to

use large amounts of expensive high resolution data, which can be hard to collect in some domains.

## 5. Conclusion

We introduced INRFlow, a flow matching generative model for continuous function spaces designed to operate directly in ambient space. Our approach dispenses with the practical complexities of training latent space generative models (Du et al., 2021a; Dupont et al., 2022a), such as the dependence on domain-specific compressors for different data domains or tuning of hyper-parameters of the data compressor (*i.e.* adversarial weight, KL term, etc.). Inspired by deterministic encoding of INRs, we introduced a conditionally independent point-wise training objective that decomposes the target vector field and allows to continuously evaluate the generated samples, similar to INRs, enabling resolution changes at inference time.

Our results on both image, 3D point cloud and protein folding benchmarks show the strong performance of INRFlow, as well as, its trivial adaption across modalities which we believe is the cornerstone of a good generative modeling architecture. In conclusion, INRFlow represents a promising direction for flow matching generative models, offering a powerful and domain-agnostic framework. Future work could explore further improvements in training efficiency applying tricks orthogonal to our contribution (Sehwag et al., 2024) and investigate co-training of multiple data domains to enable multi-modality generation in an end-to-end learning paradigm.

## Impact Statement

This paper concerns the generative modeling methodology. While we do not see immediate societal implications from our technical contribution, there are potential consequences when it is used in training foundational generative models.

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

# A. Model Configuration and Training Settings

## A.1. Image

We provide detailed model configurations and training settings of INRFlow for image generation in Tab. 6. We develop model sizes small (S), base (B), large (L), and extra large (XL) to approximately match the number of parameters in DiT (Peebles & Xie, 2023). For image experiments we implement the "psuedo" coordinate of latents as 2D grids and coordinate-value pairs are assigned to different latents based on their distances to the latent coordinates. To embed coordinates, we apply standard Fourier positional embedding (Vaswani et al., 2017) for ambient space coordinate input in both encoder and decoder. The Fourier positional embedding is also applied to the "psuedo" coordinate of latents. On image generation, we found that applying rotary positional embedding (RoPE) (Su et al., 2024) slightly improves the performance of INRFlow. Therefore, RoPE is employed for largest INRFlow-XL model. For all the models we share the following training parameters except the training_steps across different experiments. On image generation, all models are trained with batch size 256, except for INRFlow-XL reported in Tab. 2 and Tab. 3, which are trained for 1.7M steps with batch size 512.

```
default training config:
    optimizer='AdamW'
    adam_beta1=0.9
    adam_beta2=0.999
    adam_eps=1e-8
    learning_rate=1e-4
    weight_decay=0.0
    gradient_clip_norm=2.0
    ema_decay=0.999
    mixed_precision_training=bf16
```

In Tab. 7, we also compare the size of models trained on ImageNet-256, training cost (*i.e.* product of batch size and training iterations), and inference cost (*i.e.* NFE, number of function evaluation). Note that for models that achieve better performance than INRFlow, many of them are trained for more iterations. In addition, at inference time INRFlow applies simple first order Euler sampler with 100 sampling steps, which uses less NFE than many other baselines.

| Model | Layers | Hidden size | #Latents | Heads | Decoder layers | #Params |
|---|---|---|---|---|---|---|
| INRFlow-S | 12 | 384 | 1024 | 6 | 1 | 35M |
| INRFlow-B | 12 | 768 | 1024 | 12 | 1 | 138M |
| INRFlow-L | 24 | 1024 | 1024 | 16 | 1 | 458M |
| INRFlow-XL | 28 | 1152 | 1024 | 16 | 2 | 733M |

*Table 6.* Detailed configurations of INRFlow for image generation.

## A.2. Image-to-3D

For image-to-3D point cloud generation we trained INRFlow on Objaverse (Deitke et al., 2023), which contains 800k 3D objects of great variety. In particular, conditional information (i.e., an image) is integrated to our model through cross-attention. For each object in Objaverse, we sample point cloud with $16k$ points. To get images for conditioning, each object is rendered with 40 degrees field of view, $448 \times 448$ resolution, at 3.5 units on the opposite sides of $x$ and $z$ axes looking at the origin. We extract features via DINOv2 (Oquab et al., 2023) which is concatenated with Plucker ray embedding (Plucker, 2018) of each patch in DINOv2 feature. In each block, the learnable latent vector $z_{f_t}$ cross attends to image feature. During training, the image conditioning is dropped randomly with 10% probability. Therefore, our model can also benefit from popular classifier-free guidance (CFG) to increase the guidance strength. The model is trained with batch size 384 for 500k iterations. During sampling, we use an Euler-Maruyama sampler with 500 steps to generate point clouds. We train an INRFlow-XL size model of 866M parameters similar to the one reported in Tab. 11.

One particularity for image-to-3D point cloud generation is that we assign input elements to latents through a hash code, so that neighboring input elements are likely (but not certainly) to be assigned to the same latent token. We found that the improvements of spatial aware latents in 3D to not be as substantial as in the 2D image setting, so we report results with a vanilla PerceiverIO architecture for simplicity. To embed coordinates, we apply standard Fourier positional embedding (Vaswani et al., 2017) for ambient space coordinate input in both encoder and decoder.

## A.3. Protein Folding

In our experiments we use SwissProt set (Boeckmann et al., 2003) taking the ground truth structures from the AlphaFold Database (Varadi et al., 2022). We select a random set of 10k protein structures to train INRFlow. In this setting, the coordinate-value pairs represent atoms in the protein, where

| Model | # Train data | # params | bs.×it. | NFE | FID↓ | IS↑ |
|---|---|---|---|---|---|---|
| ADM (Dhariwal & Nichol, 2021) | 1.28M | 554M | 507M | 1000 | 10.94 | 100.9 |
| RIN (Jabri et al., 2023) | 1.28M | 410M | 614M | 1000 | 3.42 | 182.0 |
| HDiT (Crowson et al., 2024) | 1.28M | 557M | 742M | 100 | 3.21 | 220.6 |
| Simple Diff. (U-ViT 2B) (Hoogeboom et al., 2023) | 1.28M | 2B | 1B | - | 2.77 | 211.8 |
| DiT-XL (Peebles & Xie, 2023) | 9.23M | 675M | 1.8B | 250 | 2.27 | 278.2 |
| VDM++ (U-ViT 2B) (Kingma & Gao, 2023) | 1.28M | 2B | 1.4B | 512 | 2.12 | 267.7 |
| SiT-XL (Ma et al., 2024) | 9.23M | 675M | 1.8B | 500 | 2.06 | 270.2 |
| INRFlow-XL (ours) | 1.28M | 733M | 870M | 100 | 3.74 | 228.8 |

*Table 7.* Comparison of INRFlow and baselines in # params and training cost (*i.e.* product of batch size and training iterations). Some numbers are borrowed from (Crowson et al., 2024).

the "coordinate" part is a set of features of that particular atom. In particular, we use the atom features shown in Tab. 8 together with the eigenvectors of the graph laplacian for each residue, as previously done in (Wang et al.). In particular, we found it beneficial to also concatenate to the atomic features the amino-acid embedding of its corresponding amino-acid, which we obtain from a pre-trained ESM model (ESM-650M), a language model for protein sequences masking only on sequence data and not 3D structures.

We aggregate information into spatially-aware latents by cross-attending all atoms belonging to a particular amino-acid with its particular latent. In practice, since the number of atoms for each aminoacid type is always fixed, one might also simply use a linear layer. Finally, for this task we train a XL size model for 100k iterations with batch size 256.

## B. Performance vs Model Size

To demonstrate the scalability of INRFlow we train models of different sizes including small (S), base (B), large (L), and extra-large (XL) on ImageNet-256. We show the performance of different model sizes using FID-50K in Fig. 5(a). We observe a clear improving trend when increasing the number of parameters as well as increasing training steps. This demonstrates that scaling the total training Gflops is important to improved generative results as in other ViT-based generative models (Peebles & Xie, 2023; Ma et al., 2024).

## C. Performance vs Training Compute

We compare the performance vs total training compute of INRFlow and DiT (Peebles & Xie, 2023) in Gflops. INRFlow-linear denotes the variant of INRFlow where the cross-attention in the spatial aware encoder is replaced with grouping followed by a linear layer. We found this could be an efficient variant of standard INRFlow while still achieving competitive performance. Fig. 6 shows the comparison of the training compute in Gflops vs FID-50K between INR-Flow and latent diffusion model DiT (Peebles & Xie, 2023) including the tranining compute of the first stage VAE. We

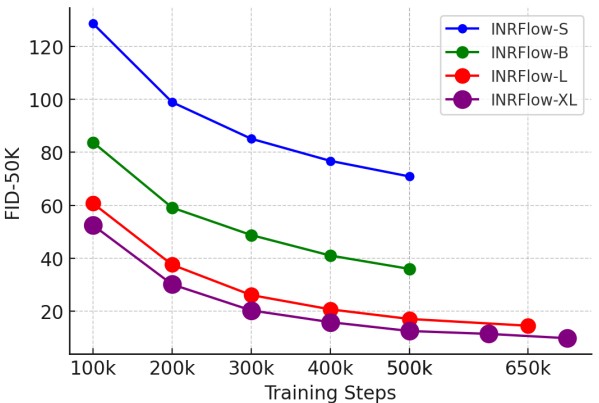

*Figure 5.* FID-50K over training iterations with different model sizes, where we see clear benefits of scaling up model sizes.

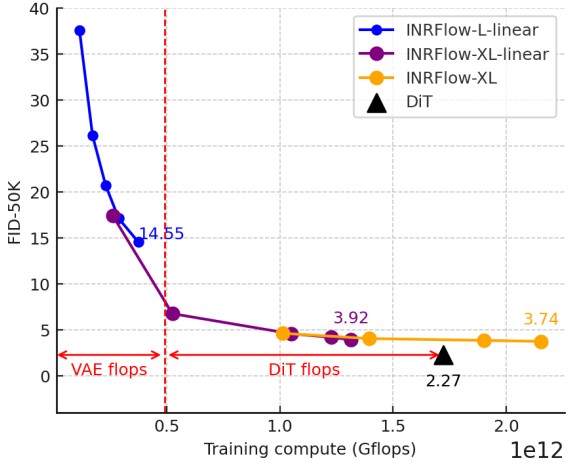

*Figure 6.* Comparing the performance vs total training compute comparison of INRFlow and DiT (Peebles & Xie, 2023).

estimate the training cost of VAE based the model card listed in HuggingFace[1]. As shown, the training cost of VAE

---

[1] https://huggingface.co/stabilityai/sd-vae-ft-mse

| Name | Description | Range |
|------|-------------|-------|
| `atomic` | Atom type | one-hot of 35 elements in dataset |
| `degree` | Number of bonded neighbors | $\{x : 0 \leq x \leq 6, x \in \mathbb{Z}\}$ |
| `charge` | Formal charge of atom | $\{x : -1 \leq x \leq 1, x \in \mathbb{Z}\}$ |
| `valence` | Implicit valence of atom | $\{x : 0 \leq x \leq 6, x \in \mathbb{Z}\}$ |
| `hybrization` | Hybrization type | $\{sp, sp^2, sp^3, sp^3d, sp^3d^2, other\}$ |
| `aromatic` | Whether on a aromatic ring | $\{True, False\}$ |
| `num_rings` | number of rings atom is in | $\{x : 0 \leq x \leq 3, x \in \mathbb{Z}\}$ |
| `ESM embedding` | Amino-acid embedding | $x \in \mathbb{R}^{1280}$ |

*Table 8.* Atomic features included in INRFlow for protein folding.

is not negligible and reasonable models with FID $\approx 6.5$ can be trained for the same cost.

Admittedly, under equivalent training Gflops, INRFlow achieves comparable but not as good performance as DiT in terms of FID score (with a difference smaller than $1.65$ FID points). We attribute this gap to the fact that DiT's VAE was trained on a dataset much larger than ImageNet, using a domain-specific architecture (*e.g.* a convolutional U-Net). We believe that the simplicity of implementing and training INRFlow models in practice, and the trivial extension to different data domains (as shown in Sect. F) are strong arguments to counter an FID difference of smaller than $1.65$ points. In addition, applying masking tricks orthogonal to our approach like the ones in (Sehwag et al., 2024) can help mitigate the training compute difference.

In addition, due to the flexibility of cross-attention decoder in INRFlow, one can easily conduct random sub-sampling to reduce the number of decoded coordinate-value pairs during training which ca also saves computation. Fig. 7 shows how number of decoded coordinate-value pairs affects the model performance as well as Gflops in training. An image of resolution 256×256 contains 65536 pixels in total which is the maximal number of coordinate-value pairs during training. As see in Fig. 5(b), a model decoding 4096 coordinate-value pairs saves more than 20% Gflops over one decoding 16384.

## D. Spatial-aware Latent

| Psuedo-coord | # latents | FID-CLIP(↓) | FID(↓) |
|--------------|-----------|-------------|--------|
| grid | 1024 | 7.32 | 9.74 |
| random | 1024 | 11.66 | 17.99 |
| KMeans++ | 1024 | 10.42 | 15.56 |
| random | 2048 | 8.95 | 11.86 |

*Table 9.* Performance of INRFlow on protein folding.

In image domain, we define pseudo coordinates to lie on a 2D grid, which results in pixels grouping as patches of same size. Tab. 9 lists the results of an ablation study on LSUN-church-256 to compare different pseudo-coordinates. We trained all models for 200K steps with batch size 128

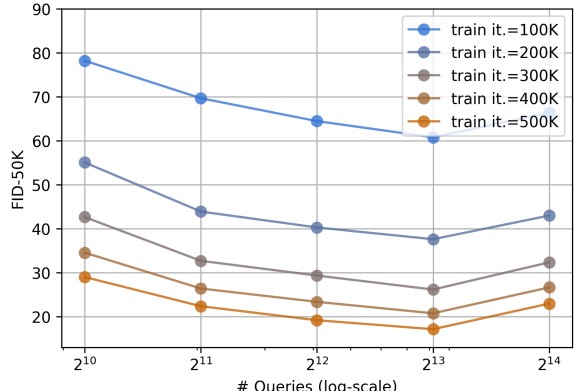

*Figure 7.* FID-50K over training iterations with different number of decoded coordinate-value pairs during training and the corresponding compute cost for a single forward pass.

and report results in the table below. INRFlow achieves the best performance when using the default grid pseudo coordinates. When using randomly sampled pseudo coordinates we observe a drop in performance.

We attribute this to the fact that when pseudo-coordinates are randomly sampled, each spatial-aware latent effectively does a different amount of work (since pixels only cross-attend to the nearest pseudo-coordinate). This unbalanced load across latents makes encoding less efficient. There are a few different ways to deal with this without necessarily relying on a grid, one is to cluster similar pseudo-coordinates to provide an equidistant distribution in 2D space (*i.e.* KMeans++ initialization), another one is to increase the number of spatial-aware latents so that each latent has to do less work. We empirically see that both of this options are effective. Ultimately, having pseudo-coordinates lie on a grid strikes a good balance of efficiency and effectiveness.

## E. Architecture Ablation

We also provide an architecture ablation in Tab. 10 showcasing different design decisions. We compare two variants

| Model | FID(↓) | Precision(↑) | Recall(↑) |
|---|---|---|---|
| PerceiverIO | 65.09 | 0.38 | 0.01 |
| INRFlow (ours) | **7.03** | **0.69** | **0.34** |

*Table 10.* Benchmarking vanilla PerceiverIO and INRFlow with spatially aware latents on LSUN-Church-256 (Yu et al., 2015).

of Transformer-based architectures INRFlow: a vanilla PerceiverIO that directly operates on ambient space, but without using spatial aware latents and INRFlow. As it can be seen, the spatially aware latents introduced in INRFlow greatly improve performance across all metrics in the image domain, justifying our design decisions. We note that we did not observe the same large benefits for 3D point clouds, which we hypothesize can be due to their irregular structure.

## F. Unconditional 3D Point Cloud generation

| Model | Layers | Hidden size | #Latents | Heads | Decoder layers | #Params |
|---|---|---|---|---|---|---|
| INRFlow-B | 9 | 512 | 1024 | 4 | 1 | 108M |
| INRFlow-L | 12 | 512 | 1024 | 4 | 1 | 204M |
| INRFlow-XL | 28 | 1152 | 1024 | 16 | 1 | 866M |

*Table 11.* Detailed configurations of INRFlow for point cloud generation.

For completeness we also tackle unconditional 3D point cloud generation on ShapeNet (Chang et al., 2015). Note that our model does not require training separate VAEs for point clouds, tuning their corresponding hyper-parameters or designing domain specific networks. We simply adapt our architecture for the change in dimensionality of coordinate-value pairs (*e.g.* $f : \mathbb{R}^2 \to \mathbb{R}^3$ for images to $f : \mathbb{R}^3 \to \mathbb{R}^3$ for 3D point clouds.). Note that for 3D point clouds, the coordinates and values are equivalent. In this setting, we compare baselines including LION (Vahdat et al., 2022) which is a recent state-of-the-art approach that models 3D point clouds using a latent diffusion type of approach. Following (Vahdat et al., 2022) we report MMD, COV and 1-NNA as metrics. To have a straightforward comparison with baselines, we train INRFlow-B with to approximately match the number of parameters as LION (Vahdat et al., 2022) (110M for LION vs 108M for INRFlow, see Tab. 11) on the same datasets (using per sample normalization as in Tab. 17 in Vahdat et al. (2022)). On ShapeNet, INRFlow models are trained for 800K iterations with a batch size of 16.

We show results for category specific models and for an unconditional model jointly trained on 55 ShapeNet categories in Tab. 12. INRFlow-B obtains strong generation results on ShapeNet despite being a domain agnostic approach and outperforms LION in most datasets and metrics. Note that INRFlow-B has comparable number of parameters and the same inference settings than LION so this is fair comparison.

Finally, we also report results for a larger model INRFlow-L (with ×2 the parameter count as LION) to investigate how INRFlow improves as with increasing model size. We observe that with increasing model size, INRFlow typically achieves better performance than the base version. This further demonstrates scalability of our model on ambient space of different data domains. More point cloud samples can be found in Appendix J.

## G. Quantitative Results on Protein Folding

This section includes the quantitative evaluation of the protein folding task. In particular, we randomly selected 512 proteins from the AFDB-SwissProt dataset (Varadi et al., 2022) and use them as a test set. We compare INRFlow with an open-source replication of AlphaFold3 (Abramson et al., 2024) (ie. Boltz-1 (Wohlwend et al., 2024)), which is the SOTA approach for protein folding. Noted that AlphaFold3 is extremely domain-specific, using complex and curated architectural designs for protein folding. For example, it relies on multi-sequence alignment and template search on existing proteins. It also designs a triangle self-attention update for atomic coordinates. Whereas INRFlow makes no assumptions about data domain and effectively models different tasks under an equivalent architecture and training objective. We report $C_\alpha$-LDDT and TM-score which are commonly used metrics to evaluate how predicted protein structures align with ground truth (Tab. 13). Results indicate that INRFlow, which uses a domain-agnostic architecture performs decently well on protein folding even when compared to SOTA models that require intricate domain-expertise embedded in the architecture. Note that we have not optimized INRFlow for hyper-parameters in the protein folding experiment.

## H. Implementation of Resolution Agnostic Generation

One can interpret the spatial-aware latents computed from INRFlow's encoder as the "latent codes that are transformed into network parameters". For INRFlow these latents codes are used to compute $v, k$ for the cross-attention block in the decoder, which takes in queries $q$ (*i.e.* coordinate-value pairs) at any resolution. The only thing that we need to do in order to obtain consistent outputs across different query resolutions is to keep the resolution of the encoder fixed, while the decoder can be queried at a different resolution. This is trivially achieved by employing a simple sub-sampling operation (*i.e.* grid sub-sampling) and keeping the sub-sampling operation fixed during inferenc (Fig. 8). This simple technique allows us to change the resolution at inference time without any other additional tricks regarding noise alignment at different resolutions and produces crisp and consistent examples at higher resolutions that the one

| Category | Model | MMD↓ | | COV↑ (%) | | 1-NNA↓ (%) | |
|---|---|---|---|---|---|---|---|
| | | CD | EMD | CD | EMD | CD | EMD |
| Airplane | ShapeGF (Cai et al., 2020) | 0.3130 | 0.6365 | **45.19** | 40.25 | 81.23 | 80.86 |
| | SP-GAN (Li et al., 2021) | 0.4035 | 0.7658 | 26.42 | 24.44 | 94.69 | 93.95 |
| | GCA (Zhang et al., 2021) | 0.3586 | 0.7651 | 38.02 | 36.30 | 88.15 | 85.93 |
| | LION (Vahdat et al., 2022) (110M) | 0.3564 | 0.5935 | 42.96 | **47.90** | 76.30 | 67.04 |
| | **INRFlow**-B (ours) (108M) | **0.2861** | 0.5156 | 43.38 | 47.54 | 75.55 | 64.95 |
| | **INRFlow**-L (ours) | 0.2880 | **0.5052** | 44.44 | 47.16 | **62.20** | **62.96** |
| Chair | ShapeGF (Cai et al., 2020) | 3.7243 | 2.3944 | 48.34 | 44.26 | 58.01 | 61.25 |
| | SP-GAN (Li et al., 2021) | 4.2084 | 2.6202 | 40.03 | 32.93 | 72.58 | 83.69 |
| | GCA (Zhang et al., 2021) | 4.4035 | 2.5820 | 45.92 | 47.89 | 64.27 | 64.50 |
| | LION (Vahdat et al., 2022) (110M) | 3.8458 | 2.3086 | 46.37 | 50.15 | 56.50 | 53.85 |
| | **INRFlow**-B (ours) (108M) | 3.6310 | **2.1725** | 46.67 | **53.31** | 55.43 | **51.13** |
| | **INRFlow**-L (ours) | **3.5145** | 2.1860 | **49.39** | 49.84 | **50.52** | 51.66 |
| Car | ShapeGF (Cai et al., 2020) | 1.0200 | 0.8239 | **44.03** | 47.16 | 61.79 | 57.24 |
| | SP-GAN (Li et al., 2021) | 1.1676 | 1.0211 | 34.94 | 31.82 | 87.36 | 85.94 |
| | GCA (Zhang et al., 2021) | 1.0744 | 0.8666 | 42.05 | 48.58 | 70.45 | 64.20 |
| | LION (Vahdat et al., 2022) (110M) | 1.0635 | 0.8075 | 42.90 | **50.85** | 59.52 | **49.29** |
| | **INRFlow**-B (ours) (108M) | 0.9923 | **0.7692** | 43.46 | 47.44 | 60.36 | 53.27 |
| | **INRFlow**-L (ours) | **0.9660** | 0.7846 | **44.03** | 48.86 | **53.83** | 54.55 |
| All (55 cat) | LION (Vahdat et al., 2022) (110M) | 3.4336 | 2.0953 | 48.00 | 52.20 | 58.25 | 57.75 |
| | **INRFlow**-B (ours) (108M) | 3.2586 | 2.1328 | 49.00 | 50.40 | 54.65 | 55.70 |
| | **INRFlow**-L (ours) | **3.1775** | **1.9794** | **49.80** | **52.39** | **51.80** | **53.90** |

*Table 12.* Generation performance metrics on Airplane, Chair, Car and all 55 categories jointly. All models were trained on the ShapeNet dataset from PointFlow (Yang et al., 2019). Both the training and testing data are normalized individually into range [-1, 1].

| Model | $C_\alpha$-LDDT(↑) | TM-Score(↑) |
|---|---|---|
| Boltz1 (Wohlwend et al., 2024) | 0.923 | 0.812 |
| INRFlow | 0.722 | 0.664 |

*Table 13.* Performance of INRFlow on protein folding.

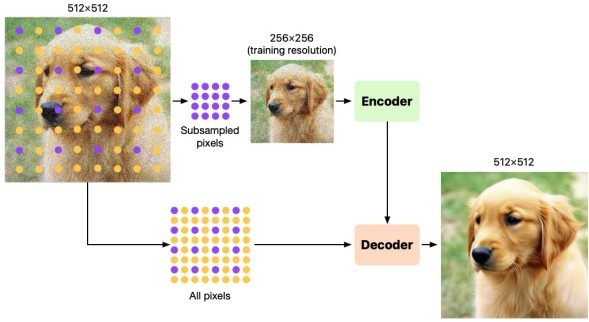

*Figure 8.* Illustration of resolution agnostic sampling. When generating higher-resolution images in inference (e.g., 512×512), 256×256 coordinate-value pairs (consistent to setting in training) are selected through grid subsampling and are fed to the encoder. The decoder takes in the full 512×512 coordinate-value pairs to predict velocities of all the pixel values. The model repeats the process in inference to generate a 512×512 image.

used in training (more results see Fig. 4 and 14).

# I. Additional ImageNet Samples

We show uncurated samples of different classes from INRFlow-XL trained on ImageNet-256 in Fig. 9 and Fig. 10. Guidance scales in CFG are set as 4.0 for loggerhead turtle, macaw, otter, coral reef and 2.0 otherwise.

# J. Additional ShapeNet Samples

We show uncurated samples from INRFlow-L trained jointly on 55 ShapeNet categories in Fig. 11.

# K. Additional Objaverse Samples

We show additional Objaverse samples from INRFlow in Fig. 12

# L. Additional SwissProt Samples

We show additional SwissProt samples from INRFlow in Fig. 13

# M. Additional Resolution Agnostic Image Samples

We show additional samples generated at different resolutions from INRFlow trained on ImageNet-256 in Fig. 14.

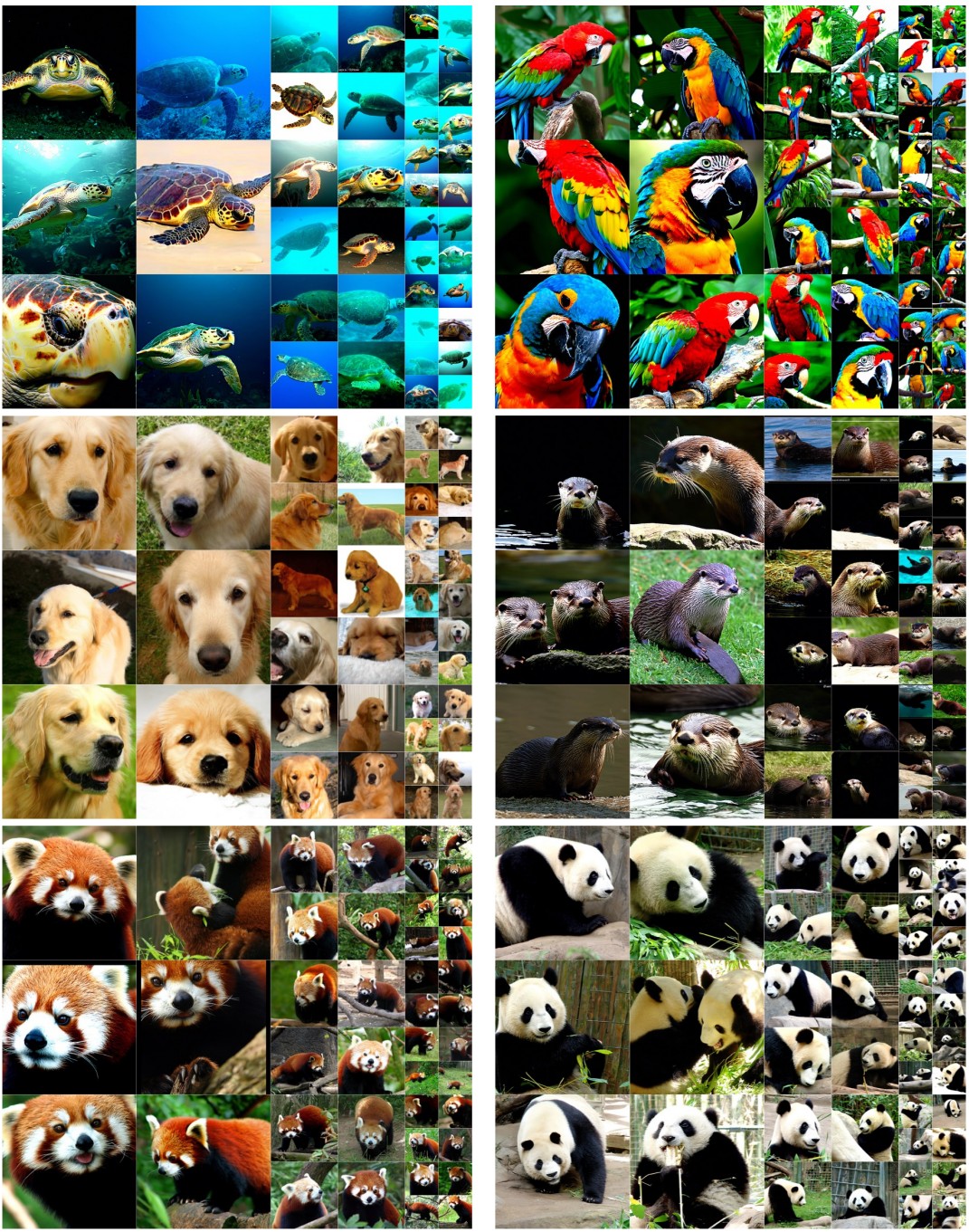

*Figure 9.* Uncurated samples of class labels: loggerhead turtle (33), macaw (88), golden retriever (207), otter (360) and red panda (387), and panda (388) from INRFlow trained on ImageNet-256.

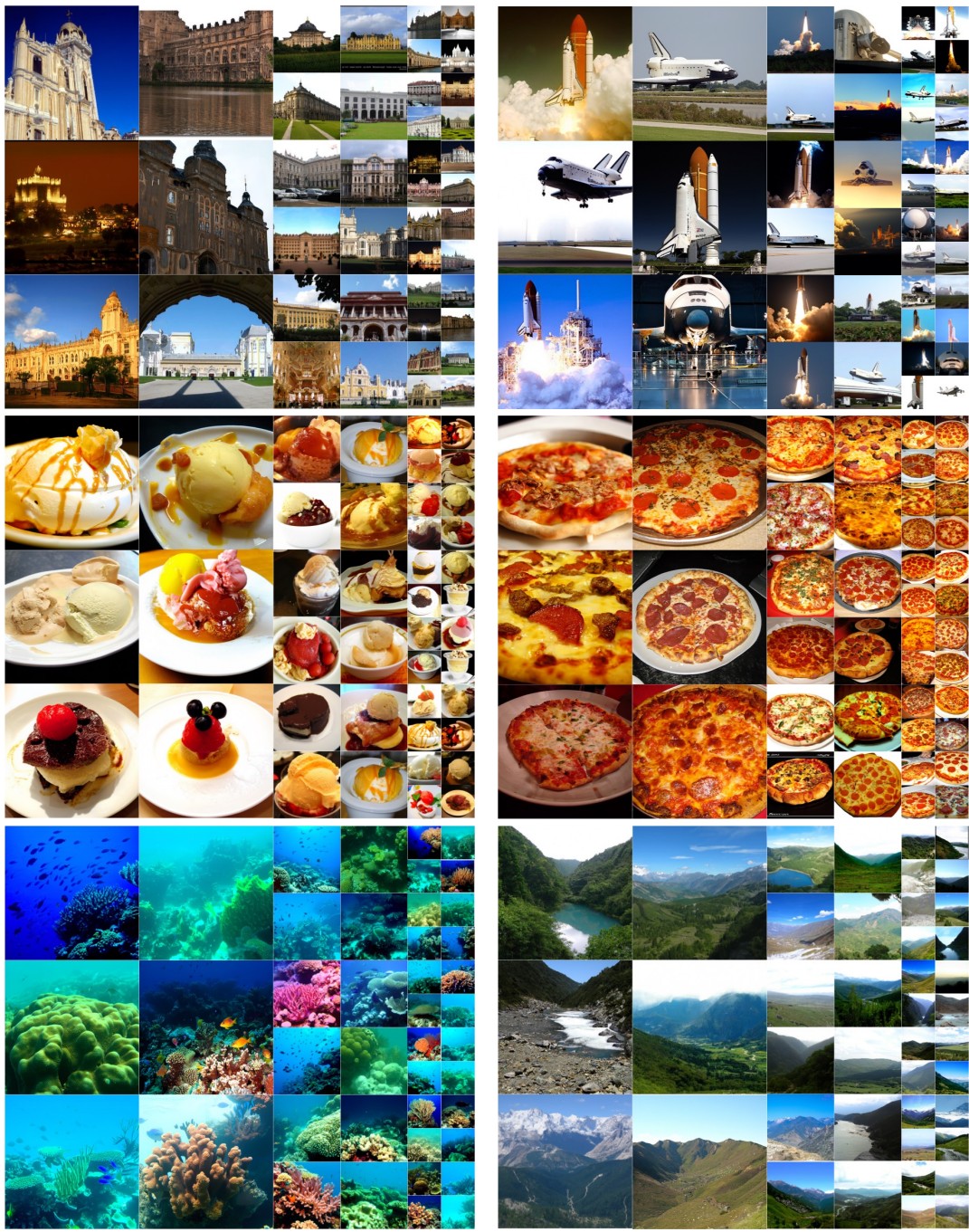

*Figure 10.* Uncurated samples of class labels: palace (698), space shuttle (812), ice cream (928), pizza (963), coral reef (973), and valley (979) from INRFlow trained on ImageNet-256.

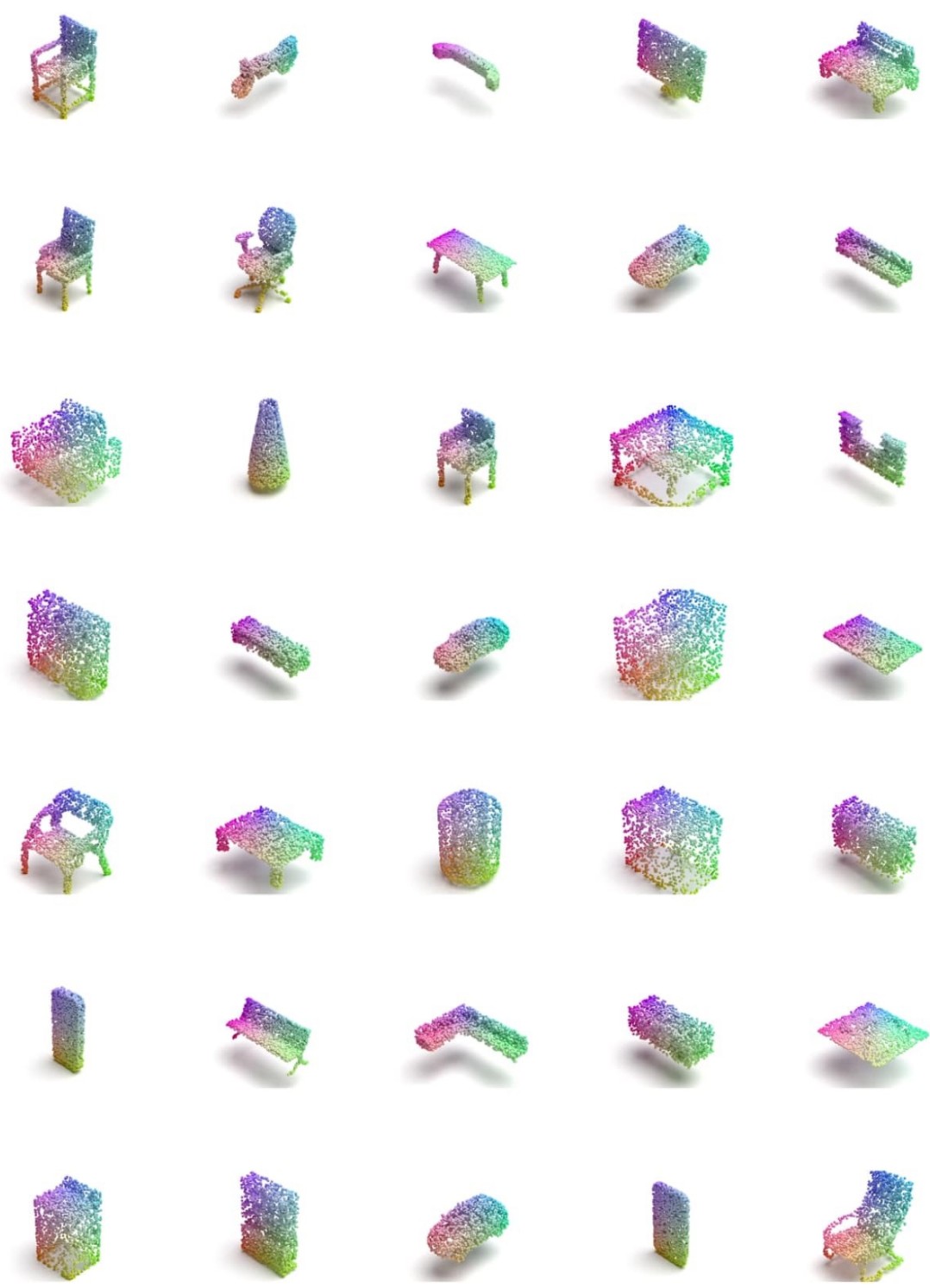

*Figure 11.* Additional uncurated ShapeNet generations using 2048 points from the unconditional model jointly trained on 55 categories

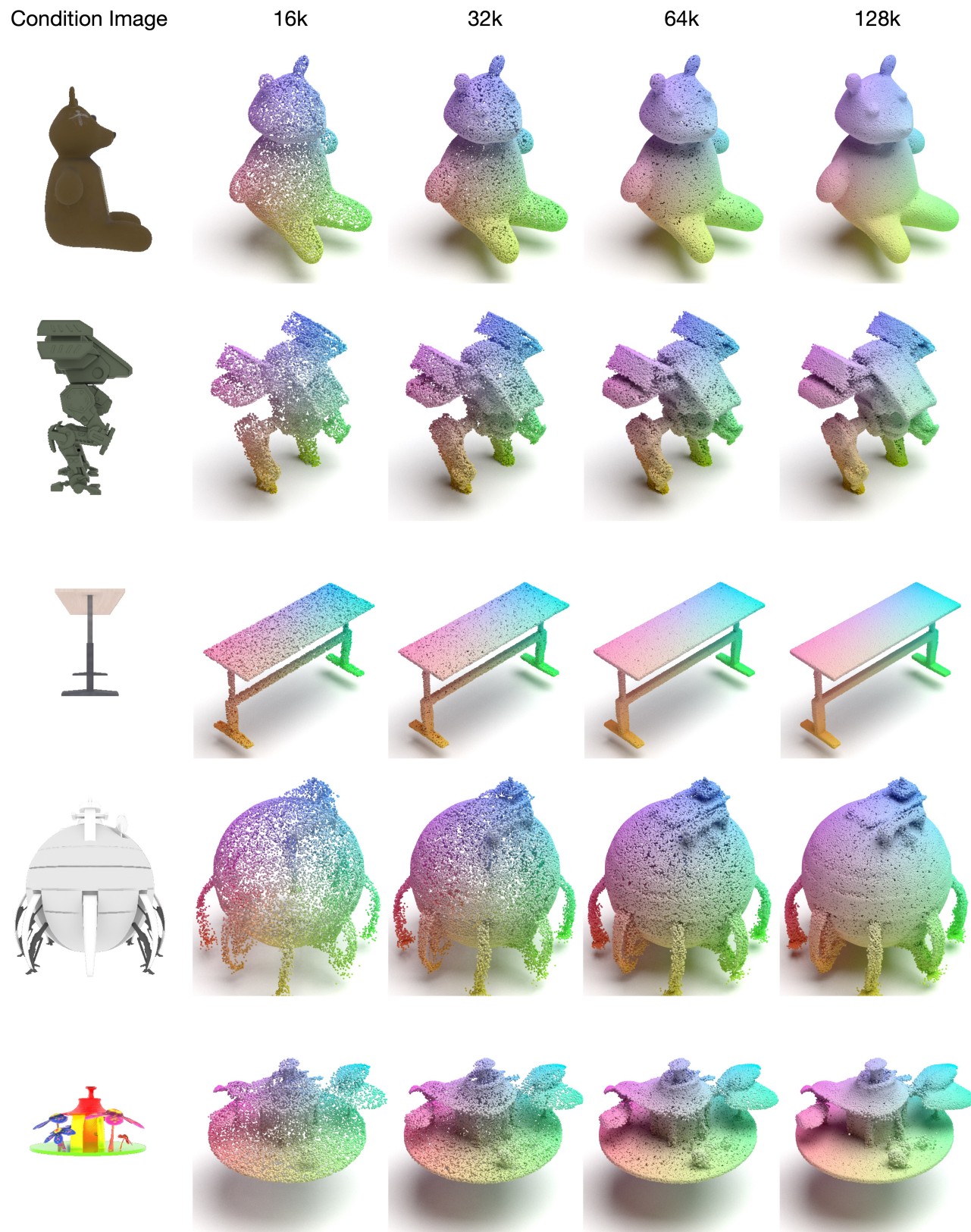

*Figure 12.* Image-conditioned point clouds with 16k, 32k, 64k, and 128k points generated from an INRFlow trained on Objaverse (training with 16k points, CFG scale 5.0).

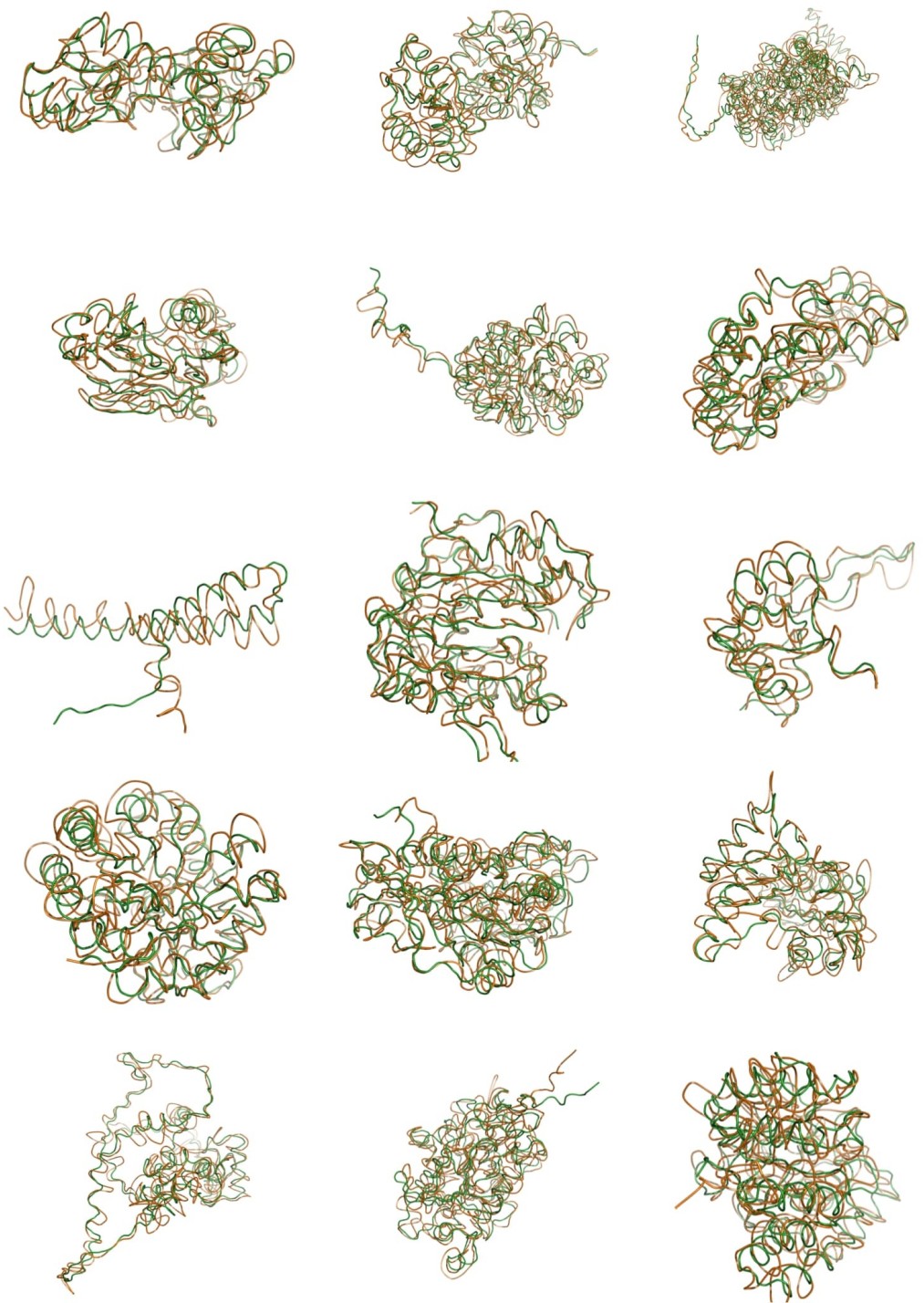

*Figure 13.* Additional examples of protein structures predicted by INRFlow on SwissProt (Boeckmann et al., 2003)

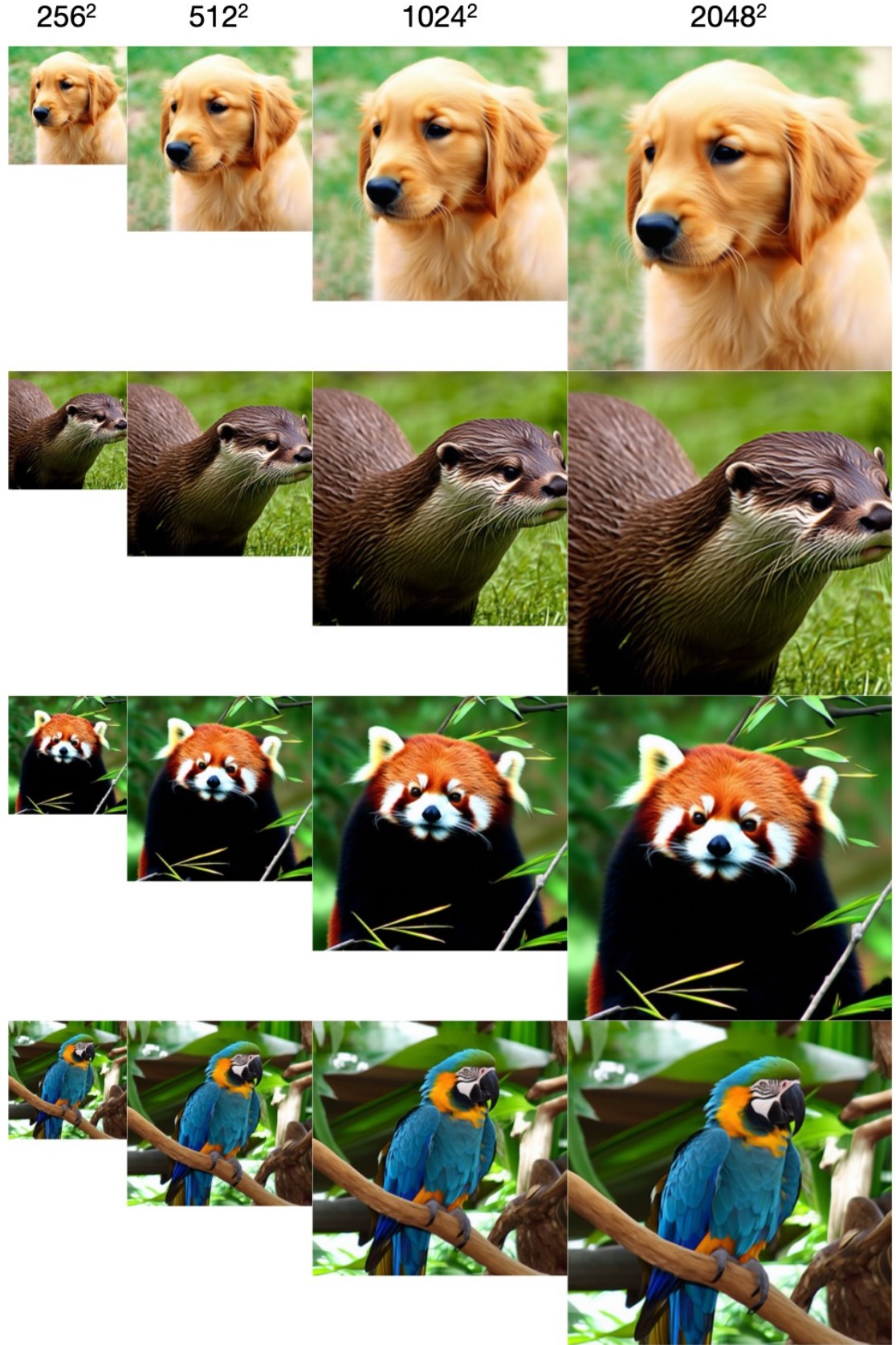

*Figure 14.* Images generated at 256, 512, 1024, and 2048 resolutions from an INRFlow trained on ImageNet-256.

