# OpenReview forum: "INRFlow: Flow Matching for INRs in Ambient Space"
_ICML.cc/2025/Conference — ICML 2025 poster_

### Official Review · Reviewer_D7hb · 2025-02-16

**Overall Recommendation:** 3

**Summary:**

Current flow matching (FM) methods are usually trained in two-stage paradigm, which sets obstacles for unifying models across data domains. To deal with this, this paper introduces INRFlow. In the proposed method, they estimate the map from coordinate to value via FM in a pointwise manner. To further model spatial dependency, they introduce latent variable via self-attention in the encoder and cross-attention in the decoder. They then applies the proposed INRFlow into image generation, image-to-3D point clound generation and protein generation.

**Claims And Evidence:**

The claims made in the submission is clear and convincing.

**Essential References Not Discussed:**

NA

**Experimental Designs Or Analyses:**

The experiement is extensive and sound.

**Methods And Evaluation Criteria:**

The goal is to train a unifying model in ambient space, using domain agnostic architecture. The proposed methods can solve the problem well. The dataset used (FFHQ-256, LSUN-Church-256, ImageNet-128/256 etc.) are diverse and standard, which shows the generality of the proposed method.

**Other Comments Or Suggestions:**

In 3.3 and 3.4, the paper use $x_{f_t}$ to denote coordinate at time t. However, the coordinate should be static, which serve as covariates for neural net approximating vector fields? If I'm correct, I would suggest use $x_f$ consistently for the entire paper (e.g. match to Figure 2)

**Other Strengths And Weaknesses:**

Strength: the idea is simple and clear. Writing is easy to follow.

Weakness: There may raise some concerns on novolties, as the proposed method is essentially combination of INR with flow matching, and using attention to handle latent variables for context information. Although the architecture is absolutely new and novel, the proposed method can solve problem well. Therefore, I would suggest borderline accept.

**Questions For Authors:**

The paper is clearly written.

**Relation To Broader Scientific Literature:**

The paper use INRs in the flow matching context with self- and cross-attention for latent variables, and the same strategies can be applied to other methods.

**Theoretical Claims:**

There's no theory part in this paper.

---

> ### Author Rebuttal · Authors · 2025-04-01
>
> 1. There may raise some concerns on novolties, as the proposed method is essentially combination of INR with flow matching, and using attention to handle latent variables for context information. Although the architecture is absolutely new and novel, the proposed method can solve problem well. Therefore, I would suggest borderline accept.
>     - We want to first thank the reviewer for acknowledging that our proposed INRFlow solves different problems with a new and novel architecture. However, we want to kindly point out that our work is more than a simple combination of INR and flow matching. Though they are previous works investigating combining INR with generative models [1, 2, 3]. Which required complex multi-stage training recipes [1,2] and were not able to scale to high resolution signals [2,3]. In fact, most of the experiments are conducted on low-resolution images like 32 $\times$ 32 and 64 $\times$ 64. Our novel architecture and training objective allows us generate images up to 2048 $\times$ 2048 resolution, which is a drastic change with respect to existing work.
>     - We want to emphasize that it’s non-trivial to obtain these results at large-scale data regime that we have. Finally, our point-wise training objective allows for efficient training via sub-sampling dense domains which previous works did not explore and allow us to tackle high-resolution data. This also enables INRFlow to do inference at arbitrary resolution in inference time without additional training (Figure 4).
>
> [1] Du, Yilun, et al. "Learning signal-agnostic manifolds of neural fields." Advances in Neural Information Processing Systems 34 (2021): 8320-8331.
>
> [2] Dupont, Emilien, et al. "From data to functa: Your data point is a function and you can treat it like one." arXiv preprint arXiv:2201.12204 (2022).
>
> [3] Zhuang, Peiye, et al. "Diffusion probabilistic fields." The Eleventh International Conference on Learning Representations. 2023.

---

### Official Review · Reviewer_RpjP · 2025-02-24

**Overall Recommendation:** 4

**Summary:**

This paper presents INRFlow, a novel domain-agnostic generative model that operates in ambient space, eliminating the need for hand-crafted data compressors in different domains. The key innovation is a conditionally independent point-wise training objective, allowing INRFlow to model continuous coordinate-value mappings across diverse data types, including images, 3D point clouds, and protein structures.

The authors claim the following contributions:
- Proposing INRFlow, a flow-matching generative transformer that works on ambient space to enable single-stage generative modeling on different data domains.
- Empirical results show that INRFlow achieves competitive performance on image and 3D point cloud generation compared with strong domain-specific baselines.
- Allowing resolution changes at inference time.

**Claims And Evidence:**

See below.

**Essential References Not Discussed:**

See below.

**Experimental Designs Or Analyses:**

See below.

**Methods And Evaluation Criteria:**

See below.

**Other Comments Or Suggestions:**

NA

**Other Strengths And Weaknesses:**

#### Pros:
- The proposed method, INRFlow, is novel, offering a simple yet effective approach that is domain-agnostic, meaning the same architecture can be applied to images, 3D point clouds, and protein structures, demonstrating its adaptability.
- The experimental evaluation is comprehensive, covering multiple domains. INRFlow performs well, achieving comparable or superior results to domain-specific architectures.
- Resolution-agnostic generation: INRFlow can produce outputs at arbitrary resolutions, providing more flexibility than traditional generative models.
- Single-stage training: The method eliminates the complexity of two-stage training pipelines, making it easier to implement and extend.

#### Cons:
- The writing quality could be improved, particularly in the method section:
    - "At a high level, our encoder network takes a set of coordinate-value pairs and encodes them to learnable latents through cross-attention. These latents are then updated through several self-attention blocks to provide the final latents." – Figure 2 does not depict cross-attention in the encoder, which may confuse readers. The description should clarify that cross-attention occurs after the self-attention blocks.
    - "Firstly, our encoder utilizes spatial aware latents where each latent is assigned a “pseudo” coordinate. Coordinate-value pairs are assigned to latents based on their distances on coordinate space." – This explanation is unclear. If the authors mean that higher resolution creates pseudo labels, further details are needed. How are these pseudo labels assigned? Interpolation? KNN? A more detailed explanation is necessary.

- An ablation study would strengthen the empirical evaluation by providing insights into key design choices. For example, investigating the impact of attended coordinates $L$, performance as a function of training sample size, and other architectural decisions would improve the paper’s clarity and robustness.

**Questions For Authors:**

1. In the flow matching framework, the objective is to map $\mathbb{R}^d \rightarrow \mathbb{R}^d$. How did you handle this in the image-to-point-cloud setup, where the mapping involves $\mathbb{R}^2 \rightarrow \mathbb{R}^3$?
2. Does INRFlow exhibit permutation equivariance? Specifically, if the input coordinates are permuted, does the output follow the same permutation?

**Relation To Broader Scientific Literature:**

See below.

**Theoretical Claims:**

See below.

---

> ### Author Rebuttal · Authors · 2025-04-01
>
> We thank the reviewer for acknowledging INRFlow’s competitive experimental performance and flexibility in inference. We also appreciate your thoughtful comments which help substantially improve the quality of our work. Please find point-by-point response to your questions below.
>
> 1. How are spatial aware latents assigned?
>     - The input coordinate-value pairs first cross attend to spatial aware latents (which are free parameters) and then self-attention is applied to the latents.  INRFlow assigns each latent with a pseudo coordinate and each coordinate-value pairs is assigned to its closest latent. For example, in image generation, we define the pseudo coordinates for latents to lie in a 2D grid by default. Thus pixels within a patch cross attend to one latent, which is located at the center of that patch. We notice that the current description can be unclear and will further clarify the spatial-aware latents definition in the final version of the paper.
>
> 2. An ablation study would strengthen the empirical evaluation by providing insights into key design choices. For example, investigating the impact of attended coordinates L, performance as a function of training sample size, and other architectural decisions would improve the paper’s clarity and robustness.
>     - We have performed an additional ablation study on selection of pseudo-coordinates for latents and the number of training samples on LSUN church 256. Results are shown in the table below (all models are trained for 200K iterations with batch size 128). We note the following and answer the reviewers questions:
>         - The process for assigning pseudo-coordinates impacts overall performance. By default, we let pseudo-coordinates lie on a 2D grid which cover the whole image. Changing that to randomly assigned psuedo-coordinates (drawn from a uniform distribution) decreases performance due to reduced image coverage, as show by the difference between rows 1 and 2 of the table.
>         - Increasing the number of latents $L$ has a positive impact on performance, even when defining pseudo-coordinates are random, as show in rows 2 and 3.
>         - Finally, we also show performance as a function of training sample size, where training INRFlow on a subset of the full dataset (e.g., 15k ($\sim$12.5%), 30k ($\sim$25%) and 60k ($\sim$50%) samples), the model achieves comparable performance to using the full dataset only using ~50% of the training data with a linear drop in performance after that. This indicates the model is capable of learning the distribution of dataset with limited data samples and generating realistic samples.
>          | Model   | pseudo coordinate | \# latents | \# Data | FID_clip | FID   |
>          | ------- | ----------------- | ---------- | ------- | -------- | ----- |
>          | INRFlow | grid              | 1024       | 126k    | 7.32     | 9.74  |
>          | INRFlow | random            | 1024       | 126k    | 11.66    | 17.99 |
>          | INRFlow | random            | 2048       | 126k    | 8.95     | 11.86 |
>          | INRFlow | grid              | 1024       | 60k     | 7.62     | 10.6  |
>          | INRFlow | grid              | 1024       | 30k     | 7.68     | 12.14 |
>          | INRFlow | grid              | 1024       | 15k     | 12.17    | 25.52 |
>
> 3. In the flow matching framework, the objective is to map $\mathbb{R}^d \rightarrow \mathbb{R}^d$. How did you handle this in the image-to-point-cloud setup, where the mapping involves $\mathbb{R}^3 \rightarrow \mathbb{R}^3$?
>     - In 3D point cloud generation, the mapping is defined as $\mathbb{R}^3 \rightarrow \mathbb{R}^3$, where the input and output share the same space. Conceptually, one can think of the signal space as the deformation of the input (ie. a deformation from a gaussian distribution in 3D to a particular object shape). Namely given a set of points in 3D space, the model learns to predict the transformation of the point cloud in the same 3D space which leads to semantically meaningful objects, conditioned on an input image.
>
> 4. Does INRFlow exhibit permutation equivariance? Specifically, if the input coordinates are permuted, does the output follow the same permutation?
>     - Yes, INRFlow preserves permutation equivariance in decoding. The decoder in INRFlow is implemented as a cross-attention block and therefore inherits the permutation equivariance property. Namely, if one permutes the query coordinate-value pairs, the decoding results will be permuted accordingly.
>     - In the encoder, INRFlow is permutation invariant. In particular, if one changes the order of input coordinate-value pairs in encoder, the assignment to spatial-aware latents stay the same which results in unchanged spatial-aware latents $z_{f_t}$. This guarantees that the learned $z_{f_t}$ well models the mapping from coordinate space to signal space.

---

### Official Review · Reviewer_GGzL · 2025-03-09

**Overall Recommendation:** 4

**Summary:**

# Update

The authors have adequately addressed my concerns, and I expect that they incorporate:

1. more explanation on how to consistency generate images at different resolutions, and
2. explanation on the difference between their proposed methods and other function-space methods I mentioned

to the final version of the paper.

I decided not to change the score as I have already given a 4.

# Old Summary

The paper presents a  model architecture and a training algorithm for generative models that can generate data in diverse domains as long as data in these domains can be represented as sets of discrete samples of a function $f: \mathcal{X} \rightarrow \mathbb{R}^d$. These domains include images ($\mathcal{X} = \mathbb{R}^2$, $d = 3$), point clouds ($\mathcal{X} = \mathbb{R}^3$ , $d = 3$), and protein structures ($\mathcal{X} =$ a set of descriptors of amino acids in a sequence, $d = 3$).

More concretely, a data item is a finite set $f = \\{ (x,y) : x \in \mathcal{X}, y \in \mathbb{R}^d \\}$, which is a "function" in the sense that, if $(x,y_1)$ and $(x,y_2)$ are members of $f$, then $y_1 = y_2$. Another way to view a data item is to say that $f = \\{ (x_1, y_1), (x_2, y_2), \dotsc, (x_N, y_N) \\}$. We may now let $\mathbf{x} = (x_1, x_2, \dotsc, x_N)$ and $\mathbb{y} = (y_1, y_2,. \dotsc, y_N)$ be tensors. A data item is thus equivalent to an ordered pair $(\mathbf{x}, \mathbf{y})$ of tensors whose first dimensions have the same size $N$.

The idea is to create a flow matching model that represents a stochastic process that transforms the data distribution to a distribution of $(\mathbf{x}, \mathbf{y})$ where $\mathbf{y}$ is standard Gaussian noise. Let $\mathbf{y}\_t = (1-t) \mathbf{y} + t\xi$ where $\xi \sim \mathcal{N}(0,I)$. We train a neural network $v\_\theta$ so that $v\_\theta(\mathbf{x}, \mathbf{y}_t, \theta)$ gives a velocity vector $\mathbf{v}$ on the trajectory that connect $\mathbf{y}_0$ (a well-formed data item) to $\mathbf{y}_1$ (a Gaussian noise tensor). This can be done by minimizing the followng loss

\begin{align*}
\mathcal{L}\_{CICFM} = E\_{\substack{t \sim \mathcal{U}[0,1],\\\\ (\mathbf{x},\mathbf{y}) \sim p_{\mathrm{data}}, \\\\ \xi \sim \mathcal{N}(0,I) }} \Big[ v_\theta(\mathbf{x}, \mathbf{y}_t, t) - u_t(\mathbf{x}, \mathbf{y}_t | \xi)  \Big]
\end{align*}

where (again) $\mathbf{y}_t = (1-t)\mathbf{y} + t\xi$, and

\begin{align*}
u_t(\mathbf{x},\mathbf{y}_t | \xi) = \frac{\xi - \mathbf{y}_t}{1 - t}.
\end{align*}

To sample a data point, one starts with a noisy data item $(\mathbf{x}, \xi)$ where $\xi \sim \mathcal{N}(0,I)$. Then, one uses the trained network above to solve the differential equation $\partial \mathbf{y}\_t / \partial t = v_\theta(\mathbf{x}, \mathbf{y}_t, t)$ with the initial condition $\mathbf{y}_1 = \xi$ and find $\mathbf{y}_0$.

The paper proposes a model architecture for $v_\theta$ so that the data size $N$ may be different among items in the dataset. Moreover, at test time, $N$ (and, to a certain extent, $\mathbf{x}$) can take on unseen values. This is done by dividing the networks into two parts: the encoder and the decoder. The encoder takes $(\mathbf{x}, \mathbf{y}_t)$ as input and produces a latent code $\mathbf{z}$. The decoder, on the other hand, operates on each "row" $(x\_i,y\_{t,i})$ of the input $(\mathbf{x}, \mathbf{y_t})$ independently, and uses $\mathbf{z}$ to model correlations between rows.

The paper claims that their method is effective in creating generating models for (1) images, (2) 3D point clouds, (3) 3D point clouds conditioned on images, and (4) protein structures. All the generative models for all domains have the same general architectures. Due to the proposed architecture, a trained model can generate images with different resolutions and point clouds with different number of points from those that are found in the training dataset.

**Claims And Evidence:**

The paper makes several important claims.

1. A single architecture can be used to model data in different domains.
2. The paper's proposed architecture and training method yield effective generative models for the domains tested.
3. Trained models can generate data at resolutions different from those used in training time.

I believe these claims are well substantiated. The variants of the same architecture are given in the Appendix, and the authors are able to reuse these architecture on different modalities. The scores that the train models achieve are competitive with baselines. Figure 4 show images with different resolutions and point clouds with different number of points.

**Essential References Not Discussed:**

I believe that the references are adequate. However, the paper may cite papers on diffusion autoencoders such as [1] because INRFlow architecture is quite similar to that of a diffusion autoencoder.

*Citation*
* [1] Preechakul et al. "Diffusion Autoencoders: Toward a Meaningful and Decodable Representation." 2022.

**Experimental Designs Or Analyses:**

I do not find issues with experimental designs and analyses.

**Methods And Evaluation Criteria:**

I believe that the datasets and the evalution metrics used are reasonable.

However, I think the paper is not clear on how it achieves "resolution agnostic generation." This is especially true on how it generates samples in Figure 4 where the images and meshes at different resolutions are similar to one another. The paper say "we simply fix the initial noise seed and incrase number of coordinate-value pairs that are evaluated." What is unclear is the process of "fixing the noise seed" as it can means several things. It can either mean (a) fixing the seed of the random number generator or (b) fixing the noise signal $\mathbf{y}_1$ at a low resolution and then deriving from it $\mathbf{y}_1$ at other resolutions.

I don't think Option (a) makes sense. For example, let's say we fix a random seed of 42. When we want to generate a (1D) image with 2 pixels, randomization might give us the following $(x,y)$ pairs.

(0.0, 0.7), (0.5, 0.4)    (Let's call this F.)

However, if we keep the same random seed and try to generate a 1D image with 4 pixels, then we might end up with the following $(x,y)$ pairs:

(0.00, 0.7), (0.25, 0.4), (0.50, 0.9), (0.75, 0.3)   (Let's call this G.)

Here, we assume that fixing a random seed would give the same sequence of random numbers. The issue is that F and G would likely to yield very different final images. A sequence that would yield an image similar to F would be an upsampled version of F, which G is clearly not one.

As a result, Option (b) seems to make more sense. However, the paper never explains how it upsamples a random noise vector to a higher resolution. As a result, it is unclear to me how the images in Figure 4 were generated, and this makes the result there not reproducible.

**Other Comments Or Suggestions:**

N/A

**Other Strengths And Weaknesses:**

I think this paper is an interesting take on a unified architecture to model multiple types of signals.

**Questions For Authors:**

Please specify how to noise vectors are initialized to generate images and meshes in Figure 4 to resolve the clarity issue I pointed out the the "Methods and Evaluations" section.

**Relation To Broader Scientific Literature:**

I think the paper should point out a significant difference between their approach and those taken by other "function space models" such as PolyINR, GEM, and GASP. For these approaches, it is possible to sample a random function $f$ and then evaluate the function mulitiple times. This is done by first sampling a latent code $z$, and then use $z$ to parameterize a neural network. For GEM and GASP, the latent codes are transformed into network parameters. For PolyINR, $z$ is turned into parameters of a generator network. Once this neural network is obtained, we can evaluate it to get the output at any input coordinates and as many input coordinates as we want. For example, if the function models an image, we can use it to generate images at $64 \times 64$, $128 \times 128$, $256 \times 256$, or any other resolutions that we want by simply feeding the function with an appropriate grid of 2D coordinates.

However, I don't think INRFlow has the above capability. It is not possible to sample a random image function and then generate the same image at resolution $64 \times 64$, $128 \times 128$, and $256 \times 256$ afterwards. This is because, to generate images at different resolutions with INRFlow, one has to start from point samples of noisy images $\\{ (x,y) \\}$ at the specified resolutions. The crucial difference here is that, for INRFlow, the noise vector $y$ has to be supplied for each input coordinate $x$. However, the approaches discussed above do not to specify $y$. To make sure that the images at different resolutions are consistent with one another, it is necessary to make sure that the noise $y$ for each resolutions are properly correlated. I believe this is harder than what is done with PolyINR, GEM, and GASP (i.e., nothing). Moreover, as pointed out earlier, the way to ensure consistency between different resolutions are not adequately explained in the paper.

**Theoretical Claims:**

I think the proposed training algorithm is correct. However, I think Equation (5)

\begin{align*}
u_t(\mathbf{x}_f, \mathbf{y}_f | \epsilon) = (1 - t)\epsilon + t \mathbf{y}_f
\end{align*}

is wrong, and I surmise this is a typo. It should have been

\begin{align*}
u\_t(\mathbf{x}\_{f\_t}, \mathbf{y}\_{f\_t} | \epsilon) = \frac{\epsilon - \mathbf{y}\_{f\_t}}{1-t}.
\end{align*}

For reference, see Equation (20) in the Lipman et al's flow matching paper [1]. Moreover, if we use the fact that $\mathbf{y}_{f_t} = (1-t)\mathbf{y}_f + t\epsilon$ as defined in Section 3.2 of othe paper, then we have that

\begin{align*}
u\_t(\mathbf{x}\_{f\_t}, \mathbf{y}\_{f\_t} | \epsilon) = \frac{\epsilon - (1-t)\mathbf{y}\_f - t\epsilon }{1-t} = \frac{1-t}{1-t}(\epsilon - \mathbf{y}\_f) = \epsilon - \mathbf{y}_f
\end{align*}

This matches with the target value $X_1 - X_0$ in Equation 1 of the rectified flow paper [2].

*Citation*
* [1] Lipman et al. "Flow Matching for Generative Modeling." 2022.
* [2] Liu et al. "Flow Straight and Fast: Learning to Generate and Transfer Data with Rectified Flow." 2022

---

> ### Author Rebuttal · Authors · 2025-04-01
>
> We thank the reviewer for their detailed comments which helped substantially improve the clarity of the submission. Please find point-by-point response to your questions below.
>
> 1. Clarification of how upsampling in Figure 4 is conducted.
>     - We agree with the reviewer that description of these experiments/implementation caused confusion and we will clarify our explanation in the final version of the paper. The resolution-free inference process is performed in the following manner. Let’s assume we have an INRFlow model trained at resolution $N=256$ for which we want to run inference at resolution $M=512$.  We start by drawing $M^2$ pixel values from the gaussian prior at timestep t=1.  We then take $N^2$ pixels from these $M^2$ pixels (eg. via simple grid sub-sampling) and feed this set of sub-sampled $N^2$ pixels to the encoder of INRFlow to compute spatial aware latents. Once these latents are computed, we feed the $M^2$ pixel coordinates and values to the decoder of INRFlow to compute cross-attention with the spatial-aware latents, which will give us $M^2$ pixels values for the next timestep. At the next timestep, we again repeat the process where we take $N^2$ pixels from the newly computed $M^2$ pixel values via sub-sampling to again feed through the encoder.  In this setting, the spatially-aware latents act as the “neural network” parameters in GEM, GASP or PolyINR. However, instead of having these parameters explicitly parametrize a neural network or a generator, we tap into them via a cross-attention block in the decoder. To summarize: the resolution of the inputs that go into the encoder does not change at inference. We just let those inputs be a sub-sample of a higher resolution noisy image. Since the decoder of INRFlow can be evaluated continuously at any resolution, we can use it to generate an image at a higher resolution that the one seen by the encoder. We will update our explanation in the paper and include a figure in the appendix with a visual explanation of the process (link to figure: https://docs.google.com/document/d/e/2PACX-1vRVsqNt103C0EFDMdvCnWF8_2XLUps3ztW4Z5Fyu-ZHMMIKrn-Rg3TR7aep3wMvnQB-1Yivb3HeTO45/pub)
>
> 2. Typo at Equation 5.
>     - We thank the reviewer for pointing out the typo, which we agree is incorrect. We will fix this in the final version of the paper
>
> 3. Missing reference: the paper may cite papers on diffusion autoencoders such as [1] because INRFlow architecture is quite similar to that of a diffusion autoencoder.
>     - We thank the reviewer for pointing out this work. We will include and discuss it in the related work section.
>
> 4. I think the paper should point out a significant difference between their approach and those taken by other "function space models" such as PolyINR, GEM, and GASP. For these approaches, it is possible to sample a random function f and then evaluate the function mulitiple times. This is done by first sampling a latent code z, and then use z to parameterize a neural network. For GEM and GASP, the latent codes are transformed into network parameters. For PolyINR, z is turned into parameters of a generator network. Once this neural network is obtained, we can evaluate it to get the output at any input coordinates and as many input coordinates as we want.
>     - The reviewer brings up a great point, which definitely needs clarification. We believe that the gap in our explanation was that one can interpret the spatial-aware latents computed from INRFlow’s encoder as the “latent codes that are transformed into network parameters” (as the reviewer explained). For INRFlow these latents codes are used to compute $k,v$ for the cross-attention block in the decoder, which takes in queries $q$ (ie. coordinate-value pairs) at any resolution.
>     - The only thing that we need to do in order to obtain consistent outputs across different query resolutions is to keep the resolution of the encoder fixed, while the decoder can be queried at a different resolution. This is trivial to achieve by employing a simple sub-sampling operation (ie. grid sub-sampling) and keeping the sub-sampling operation fixed during inference. We will update our explanation in the paper and include a figure in the appendix with a visual explanation of the process  (link to figure: https://docs.google.com/document/d/e/2PACX-1vRVsqNt103C0EFDMdvCnWF8_2XLUps3ztW4Z5Fyu-ZHMMIKrn-Rg3TR7aep3wMvnQB-1Yivb3HeTO45/pub). This simple technique allows us to change the resolution at inference time without any other additional tricks regarding noise alignment at different resolutions and produces crisp and consistent examples at higher resolutions that the one used in training (see Fig 4 and Tab 5).
>
> [1] Preechakul et al. "Diffusion Autoencoders: Toward a Meaningful and Decodable Representation." 2022.

---

### Official Review · Reviewer_EhwM · 2025-03-11

**Overall Recommendation:** 2

**Summary:**

This paper proposes INRFlow, a novel domain-agnostic approach to learn flow matching in ambient space without the need of a pretrained domain-specific encoder. INRFlow has been evaluated on three tasks: image-to-image generation, image-to-3D point cloud generation, and protein folding. The effectiveness has been demonstrated.

Strength:
1. The strength compared with SOTA baselines is highlighted.
2. The domain-agnostic flow matching is very attractive.

Weakness and Questions:
1. The key novelty -- spatial-aware latents -- is not well explained in either the main text or the appendix. How do you select the the pseudo coordinates? I can imagine the selection of these the pseudo coordinates can significantly impact the latent code quality and the generation quality. Can you do some ablation studies on this?
2. While Figure 2 shows that the conditions can be both images and 3D point clouds, the experiments only focus on images as the condition. This causes some confusion.  Is it possible to use 3D point clouds as conditions?
3. "We found that the improvements of spatial aware latents in 3D to not be as substantial as in the 2D image setting." The pseudo coordinates are in 2D or 3D? Is this caused by the selection of the pseudo coordinates?
4. Do you have some quantitative evaluation on the protein folding task?

I am willing to improve my scores based on authors' answers to my question.

**Claims And Evidence:**

Yes

**Essential References Not Discussed:**

No

**Experimental Designs Or Analyses:**

Yes. The experimental designs make sense to me.

**Methods And Evaluation Criteria:**

Yes.

**Other Comments Or Suggestions:**

See above.

**Other Strengths And Weaknesses:**

See above.

**Questions For Authors:**

See above.

**Relation To Broader Scientific Literature:**

This work is highly related to the diffusion models, flow matching, and generative AI works.

**Theoretical Claims:**

N/A

---

> ### Author Rebuttal · Authors · 2025-04-01
>
> We thank the reviewer for highlighting the strength of INRFlow as a domain-agnostic flow matching model and thoughtful comments. Please find point-by-point response to your questions below.
>
> 1. How do you select the pseudo coordinates?[...]Can you do some ablation studies on this?
>     - In image domain, we define pseudo coordinates to lie on a 2D grid, which results in pixels grouping as patches of same size. We add an ablation study on LSUN-church-256 to compare different pseudo-coordinates. We trained all models for 200K steps with batch size 128 and report results in the table below.  INRFlow achieves the best performance when using the default grid pseudo coordinates. When using randomly sampled pseudo coordinates we observe a drop in performance.
>     - We attribute this to the fact that when pseudo-coordinates are randomly sampled, each spatial-aware latent effectively does a different amount of work (since pixels only cross-attend to the nearest pseudo-coordinate). This unbalanced load across latents makes encoding less efficient. There’re a few different ways to deal with this without necessarily relying on a grid, one is to cluster similar pseudo-coordinates to provide an equidistant distribution in 2D space, another one is to increase the number of spatial-aware latents so that each latent has to do less work. We empirically see that both of this options are effective. Ultimately, having pseudo-coordinates lie on a grid strikes a good balance of efficiency and effectiveness.
> |pseudo coordinate|\# latents|FID-clip|FID|
> |-|-|-|-|
> |grid|1024|7.32|9.74|
> |random|1024|11.66|17.99|
> |KMeans++|1024|10.42|15.56|
> |random|2048|8.95|11.86|
> 2. While Fig. 2 shows that conditions can be both images and 3D point clouds[...].This causes some confusion.
>     - In Fig 2 we tried to show that both images and 3D point clouds can be modeled by INRFlow in the same way (via spatial-aware latents). We're aware how this leads to confusion and have edited the figure to make it clear (https://docs.google.com/document/d/e/2PACX-1vRVsqNt103C0EFDMdvCnWF8_2XLUps3ztW4Z5Fyu-ZHMMIKrn-Rg3TR7aep3wMvnQB-1Yivb3HeTO45/pub).
> 3. "Improvements of spatial-aware latents in 3D to not be as substantial as in the 2D image setting." [....] Is this caused by selection of the pseudo coordinates?
>     - For pointclouds, pseudo coordinates are defined in 3D space as opposed to in 2D for images. There’re two main reasons that we found for 3D pointclouds not benefiting from spatial-aware latents:
>         - Pointclouds are sparse representation of 3D geometry. If we define a fixed set of pseudo-coordinates as a 3D grid, a substantial amount of latents will encode empty space, which is not efficient for encoding (similar to voxel representations). Therefore, we opt for vanilla PerceiverIO for 3D point cloud generation obtaining great results without sacrificing efficiency.
>         - In 3D, we learn a deformation field from a prior 3D pointcloud distribution (eg. a 3D gaussian) towards a training set of shapes. Given that the geometry of the pointcloud changes at different timesteps and for different samples, it’s not trivial to define a fixed set of pseudo-coordinates to effectively cover the full geometry. We believe that timestep-aware and sample-aware pseudo-coordinates are very interesting to study in the future and are part of future work.
> 4. Quantitative evaluation on protein folding task
>     - We have included a quantitative evaluation of the protein folding task. In particular, we randomly selected 512 proteins from the AFDB-SwissProt dataset and use them as a test set. We compare INRFlow with an open-source replication of AlphaFold3 [1] (ie. Boltz-1 [2]), which is the SOTA approach for protein folding. We note that AlphaFold3 is extremely domain-specific, using complex and curated architectural designs for protein folding. For example, it relies on multi-sequence alignment and template search on existing proteins. It also designs a triangle self-attention update for atomic coordinates. Whereas INRFlow makes no assumptions about data domain and effectively models different tasks under an equivalent architecture and training objective.
>     - We report $C_\alpha$-LDDT and TM-score (higher the better) which are commonly used metrics to evaluate how predicted protein structures align with ground truth. Results indicate that INRFlow, which uses a domain-agnostic architecture performs decently well on protein folding even when compared to SOTA models that require intricate domain-expertise embedded in the architecture. Note that we have not optimized INRFlow for model size or other hyper-parameters in the protein folding experiment.
> |Model|$C_\alpha$-LDDT|TM-score|
> |-|-|-|
> |AlphaFold3 (Boltz-1)|0.923|0.812|
> |INRFlow|0.722|0.664|
>
> [1] Abramson, J., et al. "Accurate structure prediction of biomolecular interactions with AlphaFold 3." Nature (2024).
>
> [2] Wohlwend, J., et al. "Boltz-1: Democratizing Biomolecular Interaction Modeling." bioRxiv (2024).

---

### Decision · Program_Chairs · 2025-05-01

**Decision:**

Accept (poster)

**Comment:**

The proposed approach solves an important problem of domain agnostic generative model architecture. The results indicate strong performance. The reviewers recognise the value of the contribution.
One reviewer's recommendation stays bellow accept. However, this reviewer remained unresponsive during the rebuttal phase.